# Predicting Label Distribution from Multi-label Ranking

**Yunan Lu, Xiuyi Jia**[*]
School of Computer Science and Engineering
Nanjing University of Science and Technology, Nanjing 210094, China
`{luyn, jiaxy}@njust.edu.cn`

## Abstract

Label distribution can provide richer information about label polysemy than logical labels in multi-label learning. There are currently two strategies including LDL (label distribution learning) and LE (label enhancement) to predict label distributions. LDL requires experts to annotate instances with label distributions and learn a predictive mapping on such a training set. LE requires experts to annotate instances with logical labels and generates label distributions from them. However, LDL requires costly annotation, and the performance of the LE is unstable. In this paper, we study the problem of predicting label distribution from multi-label ranking which is a compromise w.r.t. annotation cost but has good guarantees for performance. On the one hand, we theoretically investigate the relation between multi-label ranking and label distribution. We define the notion of EAE (expected approximation error) to quantify the quality of an annotation, give the bounds of EAE for multi-label ranking, and derive the optimal range of label distribution corresponding to a particular multi-label ranking. On the other hand, we propose a framework of label distribution predicting from multi-label ranking via conditional Dirichlet mixtures. This framework integrates the processes of recovering and learning label distributions end-to-end and allows us to easily encode our knowledge about current tasks by a scoring function. Finally, we implement extensive experiments to validate our proposal.

## 1 Introduction

The label polysemy problem has been a popular research topic in machine learning area, in which an instance is described by multiple labels simultaneously. MLL (multi-label learning) [23] deals with label polysemy by assigning a vector with logical values to the instance, in which each logical value indicates whether the corresponding label is associated with the instance. However, MLL only gives which labels can describe the instance, but cannot directly answer a question with more polysemy, i.e., how much does each label describe the instance. Hence, label distribution [4], a real-valued vector that explicitly gives the description degrees of labels to an instance, is introduced to answer this question. Obviously, label distribution provides richer information about label polysemy than logical labels, and it has been applied in many practical application scenarios, such as sentiment analysis [14, 35, 39], facial age estimation [3, 6, 30], and so on.

There are two main methods for obtaining label distribution to characterize label polysemy. The first type is LDL (label distribution learning) [4], that is, learning a predictive mapping from a feature vector to a label distribution. LDL requires experts to directly annotate the instances with label distributions as a training set. Such methods focus on how to design a well performed LDL algorithm. Typical works include some algorithms [10, 20, 25, 37, 38] that improve LDL performance by mining

---

[*]Corresponding author

36th Conference on Neural Information Processing Systems (NeurIPS 2022).

label correlations, and some [21, 31] that improve the ability to fit complex label distributions by introducing more flexible models. The second type is LE (label enhancement) [34], that is, generating label distributions from a vector of logical labels. LE requires experts to annotate the instances with logical labels first, and then recover the label distribution by analyzing the features and labels of the training instances, finally train a predictive model by these recovered label distributions. In other words, LE can be regarded as the pre-processing of LDL to obtain the label distribution. Such methods focus on how to recover the true label distribution accurately from the given logical labels. For example, most LE algorithms [12, 22, 29, 34, 40] consider recovering more accurate label distributions by mining sample and label correlations.

Since LDL methods are trained directly on instances with true label distributions, they usually produce better performance. However, annotating instances with label distributions is costly and even impractical in some cases [34]. In contrast, LE methods only require experts to annotate the instance with logical labels, thus reducing the annotation cost. However, there is no reliable theory to guarantee that the label distribution recovered from logical labels converges to the true label distribution. In terms of the annotation form, two labels taking the same logical value are indistinguishable in the MLL case, while the label distribution usually characterizes them as labels with different description degrees. In terms of solution, logical labels provide a large solution space for the label distribution (e.g., the label distribution component corresponding to a label with a logical value of 1 can take any real value from 0 to 1), making the solution unstable and inaccurate. Therefore, we propose a hypothesis that the label distribution recovered from logical labels is not guaranteed to have the same label ranking as the true label distribution, while accurate label ranking is the key to recovering and predicting an accurate label distribution [13, 26, 27].

Fortunately, multi-label ranking [1, 2, 7] is a good annotation form to address the above problems. Multi-label ranking[2] requires experts to give which labels are relevant to the instance and further to give the ranking (strict order) of these relevant labels. Although the annotation cost of multi-label ranking is slightly higer than that of logical labels, it guarantees a ranking consistent with the true label distribution and constrains the approximated label distribution in a narrow solution space. Hence, in this paper, we investigate the problem of predicting label distribution from multi-label ranking.

On the one hand, we theoretically investigate the relation between multi-label rankings and label description vectors[3] (unnormalized label distributions). We define the notion of EAE (expected approximation error) to quantify the quality of annotation w.r.t. recovering the true label description vector; we derive the EAE for multi-label ranking and logical labels to clarify the advantage of multi-label ranking; we give the bounds of EAE for multi-label ranking, and derive the optimal range of label description vector corresponding to a particular multi-label ranking.

On the other hand, we propose a generic framework named DRAM (label **D**istribution predicting from multi-label **RA**nking via conditional Dirichlet **M**ixtures). This framework first forms a semi-adaptive prior $p^\star(\boldsymbol{d})$ for the label distribution via a scoring function with a predefined functional form and adaptive parameters, then models the predictive distribution $p(\boldsymbol{d}|\boldsymbol{x})$ by conditional Dirichlet mixtures, finally learns the model parameters by minimizing the cross-entropy of $p(\boldsymbol{d}|\boldsymbol{x})$ relative to $p^\star(\boldsymbol{d})$. This framework has two merits: 1) It allows us to flexibly encode our prior knowledge about the tasks by a scoring function, and 2) it integrates the processes of recovering and learning label distributions end-to-end. Besides, we design a comparison method whose main idea is to transform the dataset with multi-label rankings into the dataset with logical labels such that any existing LE method can be borrowed. Finally, to validate our proposal, we conduct experiments on reduced LDL datasets and a new real-world dataset that we create directly according to the task. The experimental results show that our method significantly outperforms the comparison methods and also outperforms the LDL methods directly trained on the examples with true label distribution in most cases.

---

[2]In some literature, "multi-label ranking" is a learning task; in this paper, it only denotes an annotation form.

[3]In general, label distribution $\boldsymbol{d}$ satisfies that each element $d_i \in [0, 1]$ and $\sum d_i = 1$. For simplicity, we sometimes do not consider $\sum d_i = 1$. We call such an unnormalized label distribution a label description vector.

## 2 Theoretical analysis

### 2.1 Preliminary

Let $x$ denote the feature vector of the instance and $\mathcal{Y} = \{y_i\}_{i=1}^M$ denote the label set. The label description vector $z$ is the expert's internal view of how much does each label describe the instance; $z_i \in [0, 1]$ indicates the description degree of $y_i$ to $x$. If the expert is asked to annotate the instance with logical labels, the internal label description vector will be degenerately expressed as a logical label vector $l \in \{0, 1\}^M$; the element $l_i = 1$ (or $l_i = 0$) in $l$ means that the label $y_i$ is the relevant (or irrelevant) to the instance $x$. Let $m$ denote the number of relevant labels. If the expert is asked to annotate the instance with a multi-label ranking, the internal label description vector will be degenerately expressed as a permutation $\sigma$ (which represents a total strict order) on the relevant labels; $\sigma_i$ indicates that the label $y_{\sigma_i}$ is at the $i$-th position in ascending order of the description degree; for $j \in [M] \backslash \sigma$, the label $y_j$ is an irrelevant label for $x$, where $[M] \triangleq \{1, 2, \cdots, M\}$.

Since both logical labels and multi-label ranking are reduced versions of the internal label description vector, there is consistency between them, which can be described in the following two assumptions:

**Assumption 1** *If the expert's internal label description vector $z$ is expressed as a logical label vector $l$, and $\delta > 0$ is the minimum margin[4] of label description degrees, then we have $z \in \mathcal{S}_l^\delta$, where $\mathcal{S}_l^\delta = \{z \mid (\forall l_i = 0, z_i = 0) \wedge (\forall l_j = 1, \delta \le z_j \le 1)\}$.*

**Assumption 2** *If the expert's internal label description vector $z$ is expressed as a multi-label ranking $\sigma$, and $\delta > 0$ is the minimum margin of label description degrees, then we have $z \in \mathcal{S}_\sigma^\delta$, where $\mathcal{S}_\sigma^\delta = \{z \mid (\forall i \in \sigma, \delta \le z_i \le 1) \wedge (\forall i \in [m-1], z_{\sigma_i} \le z_{\sigma_{i+1}} - \delta) \wedge (\forall i \in [M] \backslash \sigma, z_i = 0)\}$.*

Note that the margin $\delta$ is an implicit variable in the annotation process, which is not explicitly indicated by the annotation results. Therefore, the range of internal label description vector (e.g. $\mathcal{S}_\sigma^\delta$ and $\mathcal{S}_l^\delta$) is also implicit, so the range determined by the implicit interval $\delta$ is called the implicit range. Although $\delta$ is implicit, we can predefine an explicit margin $\hat{\delta}$; the range determined by the explicit margin (e.g. $\mathcal{S}_\sigma^{\hat{\delta}}$ and $\mathcal{S}_l^{\hat{\delta}}$) is called the explicit range. We are then able to generate a label description vector from the explicit range to approximate the internal label description vector.

### 2.2 Theoretical results

We first define EAE to measure the quality of a certain annotation form w.r.t. approximating the internal label description vector.

**Definition 1** *Suppose that an instance is annotated with an annotation $r$; $\delta$ and $\hat{\delta}$ are implicit and explicit margins, respectively; $\mathcal{S}_r^\delta$ and $\mathcal{S}_r^{\hat{\delta}}$ are the implicit and explicit ranges, respectively. Then the expected approximation error of $r$ to the internal label description vector is*

$$\epsilon_r^{\delta, \hat{\delta}} = \int_{z \in \mathcal{S}_r^\delta} \int_{\hat{z} \in \mathcal{S}_r^{\hat{\delta}}} \frac{1}{V_r^{\hat{\delta}} V_r^\delta} \|z - \hat{z}\|_2^2 \mathrm{d}z \mathrm{d}\hat{z}, \quad V_r^\delta = \int_{z \in \mathcal{S}_r^\delta} \mathrm{d}z, \quad V_r^{\hat{\delta}} = \int_{z \in \mathcal{S}_r^{\hat{\delta}}} \mathrm{d}\hat{z}. \quad (1)$$

Eq. (1) is essentially derived from the expectation of the squared Euclidean distance between $z$ and $\hat{z}$, i.e., $\mathbb{E}_{z, \hat{z}} \left[ \|z - \hat{z}\|_2^2 \right]$, which measures the average distance between the estimated label description vector and the internal one given the ranges $\mathcal{S}_r^\delta$ and $\mathcal{S}_r^{\hat{\delta}}$. Since $z$ and $\hat{z}$ are independent, $p(z, \hat{z}) = p(z)p(\hat{z})$. Besides, we do not consider additional assumptions to reduce the uncertainty of $z$ and $\hat{z}$; we assume that $z$ and $\hat{z}$ follow the uniform distributions on $\mathcal{S}_r^\delta$ and $\mathcal{S}_r^{\hat{\delta}}$, i.e., $p(z)p(\hat{z}) = (V_r^\delta V_r^{\hat{\delta}})^{-1}$, and derive the Eq. (1). Next, we give the EAE of multi-label ranking.

**Theorem 1** *If an instance is annotated by a multi-label ranking $\sigma$, $m$ is the number of relevant labels, $\delta$ and $\hat{\delta}$ are the implicit and explicit margins, respectively, then the EAE of $\sigma$ is*

$$\epsilon_\sigma^{\delta, \hat{\delta}} = \frac{m}{6(m+1)} \Big( (m+1)^2 (\delta^2 + \hat{\delta}^2) - 2m(\delta + \hat{\delta}) - (4m+2)\delta\hat{\delta} + 2 \Big). \quad (2)$$

---

[4]Note that it is hard for an expert to rank a set of labels with close description degrees. We hence introduce the minimum margin $\delta$ to represent the smallest difference of description degrees required for the expert to distinguish and rank these labels.

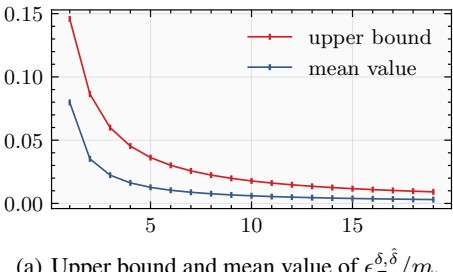

(a) Upper bound and mean value of $\epsilon_{\boldsymbol{\sigma}}^{\delta,\hat{\delta}}/m$.

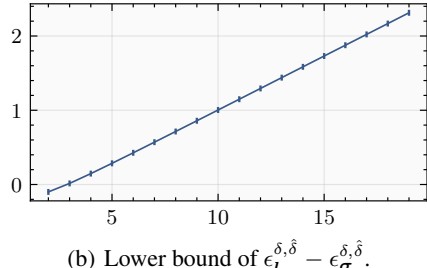

(b) Lower bound of $\epsilon_{\boldsymbol{l}}^{\delta,\hat{\delta}} - \epsilon_{\boldsymbol{\sigma}}^{\delta,\hat{\delta}}$.

Figure 1: Visualization of corollaries. The horizontal coordinates of the two figures above indicate the number of relevant labels $m$.

Theorem 1 can be proved by mathematical induction, and the details can be found in the appendix. Before giving more corollaries, we need to specify the range of the margins, i.e., $\delta$ and $\hat{\delta}$.

**Lemma 1** *If an instance is annotated by a multi-label ranking $\boldsymbol{\sigma}$, then the margins $\delta$ and $\hat{\delta}$ satisfy that $0 \leq \delta \leq m^{-1}$ and $0 \leq \hat{\delta} \leq m^{-1}$.*

Next we give some interesting corollaries to understand Theorem 1.

**Corollary 1** *If an instance is annotated by a multi-label ranking $\boldsymbol{\sigma}$, $m$ is the number of relevant labels, the explicit margin $\hat{\delta}^{\star}$ minimizing the EAE of $\boldsymbol{\sigma}$ is $\hat{\delta}^{\star} = ((2m+1)\delta + m)(m+1)^{-2}$.*

It is clear that the optimal explicit margin $\hat{\delta}^{\star}$ depends on the implicit margin $\delta$; hence we cannot obtain an exact optimum for $\hat{\delta}^{\star}$. Nevertheless, Corollary 1 helps us to narrow down the range of the optimal explicit margin considerably, i.e., $m(m+1)^{-2} \leq \hat{\delta}^{\star} \leq m^{-1}$.

**Corollary 2** *If an instance is annotated by a multi-label ranking $\boldsymbol{\sigma}$, $m$ is the number of relevant labels, $0 \leq \delta \leq m^{-1}$, $m(m+1)^{-2} \leq \hat{\delta} \leq m^{-1}$, then the EAE of $\boldsymbol{\sigma}$ is bounded by:*

$$0 \leq \epsilon_{\boldsymbol{\sigma}}^{\delta,\hat{\delta}} \leq \frac{m(m^2 + 4m + 2)}{6(m+1)^3} < \frac{1}{5}. \tag{3}$$

Corollary 2 gives $m$-dependent bounds on the EAE of the multi-label ranking. See the appendix for details of the proof.

**Corollary 3** *If an instance is annotated by a multi-label ranking $\boldsymbol{\sigma}$, $m$ is the number of relevant labels, $\delta$ and $\hat{\delta}$ are uniform over $\left[0, m^{-1}\right]$ and $\left[m(m+1)^{-2}, m^{-1}\right]$, respectively, then we have:*

$$\underset{\delta,\hat{\delta}}{\mathbb{E}} \left[ \epsilon_{\boldsymbol{\sigma}}^{\delta,\hat{\delta}} \right] = \frac{2m^4 + 8m^3 + 8m^2 + 4m + 1}{36m(m+1)^3}. \tag{4}$$

Corollary 3 can be obtained by a simple integral calculation, as detailed in the appendix. In Fig. 1(a), we visualize the expected value in Corollary 3 and the upper bound in Corollary 2. It is obvious that we can obtain a significant performance gain by ranking a small number of labels.

**Theorem 2** *If an instance is annotated by a logical label vector $\boldsymbol{l}$, $m$ is the number of relevant labels, $\delta$ and $\hat{\delta}$ are the implicit and explicit margins, respectively, then the EAE of $\boldsymbol{l}$ is*

$$\epsilon_{\boldsymbol{l}}^{\delta,\hat{\delta}} = \frac{m}{6}(2\delta^2 + 2\hat{\delta}^2 - \delta - \hat{\delta} - 3\delta\hat{\delta} + 1). \tag{5}$$

Theorem 2 gives the EAE of the logical label vector, and the proof is detailed in the appendix.

**Corollary 4** *Suppose that $\epsilon_{\boldsymbol{l}}^{\delta,\hat{\delta}_{\boldsymbol{l}}}$ and $\epsilon_{\boldsymbol{\sigma}}^{\delta,\hat{\delta}_{\boldsymbol{\sigma}}}$ are the EAE of the logical label vector $\boldsymbol{l}$ and the EAE of the multi-label ranking $\boldsymbol{\sigma}$, respectively, we have the following inequality holds for $m \geq 3$:*

$$
\begin{aligned}
\epsilon_{\boldsymbol{l}}^{\delta,\hat{\delta}_{\boldsymbol{l}}} - \epsilon_{\boldsymbol{\sigma}}^{\delta,\hat{\delta}_{\boldsymbol{\sigma}}} &\geq \frac{7m}{48}(\delta^2 - 2\delta) + \frac{m(m-1)(7m^2 + 20m + 9)}{48(m+1)^3} \\
&> \frac{7m^5 - m^4 - 46m^3 - 30m^2 + 7m + 7}{48m(m+1)^3} > 0.
\end{aligned}
\tag{6}
$$

Corollary 4 shows the advantage of the multi-label ranking over the logical labels w.r.t. approximating the true label description vector. It is obvious that as the number of relevant labels $m$ increases, $\epsilon_{\boldsymbol{l}}^{\delta,\hat{\delta}_{\boldsymbol{l}}} - \epsilon_{\boldsymbol{\sigma}}^{\delta,\hat{\delta}_{\boldsymbol{\sigma}}}$ increases at least at the rate of $\mathcal{O}(m)$, which is visualized in Fig. 1(b).

# 3 Algorithms

In this section, we consider how to train a model on the dataset $\{(\boldsymbol{x}_n, \boldsymbol{\sigma}_n)\}_{n=1}^N$ for predicting label distributions. We propose a framework DRAM to deal with this problem. Besides, we also design a comparison method.

## 3.1 DRAM framework

First, we describe how to enhance multi-label rankings into label distributions. Then, we formally give the predictive model. Finally, we derive a generic EM algorithm for our framework that works for any instantiation of the basic models and consider a concrete instantiation.

### 3.1.1 Recovering label distributions from multi-label rankings

In order to learn a mapping from instance features to label distributions, we consider enhancing multi-label rankings to label distributions. The process of enhancing the simple label (e.g., multi-label ranking and logical label) into the label distribution can be viewed as selecting label distributions that satisfy a predefined prior assumption from those consistent with simple labels. For example, some algorithms [9, 34] select those label distributions that satisfy the smoothness assumption [24]; In fact, according to the no-free-lunch axiom, no prior assumption can work for all tasks; hence we do not consider a concrete assumption in our framework. We provide a semi-adaptive scoring function $\phi(\boldsymbol{d};\boldsymbol{\theta})$ such that any assumption can be easily encoded. The scoring function allows us to build a prior distribution of label distribution $p^\star(\boldsymbol{d})$:

$$
p^\star(\boldsymbol{d}) = \frac{1}{Z_{p^\star}} \phi(\boldsymbol{d};\boldsymbol{\theta}) \int_0^\infty \mathbb{I}(t\boldsymbol{d} \in \mathcal{S}_{\boldsymbol{\sigma}}^{\hat{\delta}})\mathrm{d}t,
\tag{7}
$$

where $\mathcal{S}_{\boldsymbol{\sigma}}^{\hat{\delta}}$ is the explicit range of $\boldsymbol{\sigma}$, $Z_{p^\star}$ is a normalization constant, and $\mathbb{I}(\cdot)$ denotes the indicator function. The integral term in Eq. (7) intuitively indicates how many label description vectors can be normalized to $\boldsymbol{d}$. According to Corollary 1, we set $\hat{\delta}$ corresponding to the example $(\boldsymbol{x}_n, \boldsymbol{\sigma}_n)$ as $|\boldsymbol{\sigma}_n|(|\boldsymbol{\sigma}_n| + 1)^{-2}$. The functional form of $\phi(\boldsymbol{d};\boldsymbol{\theta})$ is predefined and the parameters can be learned adaptively. For example, we can predefine $\phi(\boldsymbol{d};\boldsymbol{\theta})$ as a Gaussian likelihood function, and leave its mean and variance to be learned.

### 3.1.2 Predictive model: conditional Dirichlet mixtures

Here we need to determine the distribution form of $\boldsymbol{d}$ conditioned on $\boldsymbol{x}$. We use Dirichlet distribution to model $\boldsymbol{d}|\boldsymbol{x}$. Since $\phi$ is any non-negative real-valued function, the prior distribution of $\boldsymbol{d}$ is usually multimodal. Therefore, we model $p(\boldsymbol{d}|\boldsymbol{x})$ with the mixture of Dirichlet distributions:

$$
p(\boldsymbol{d}|\boldsymbol{x}) = \sum_{k=1}^K f_k(\boldsymbol{x};\boldsymbol{\alpha})\mathrm{Dir}(\boldsymbol{d}|f(\boldsymbol{x};\boldsymbol{\beta}^k)),
\tag{8}
$$

where $\boldsymbol{\alpha}, \boldsymbol{\beta}^1, \cdots, \boldsymbol{\beta}^K$ are learnable parameters; $f(\boldsymbol{x};\boldsymbol{\alpha})$ outputs a $K$-dimensional positive real-valued vector with a sum of 1, and $f_k(\boldsymbol{x};\boldsymbol{\alpha})$ is its $k$-th value; $f(\boldsymbol{x};\boldsymbol{\beta}^k)$ outputs a $M$-dimensional positive real-valued vector; $\mathrm{Dir}(\cdot)$ denotes Dirichlet distribution whose details are in the appendix.

---

**Algorithm 1** Generic DRAM

---

**Require:** training set $\{(\boldsymbol{x}_n, \boldsymbol{\sigma}_n)\}_{n=1}^N$, testing instance $\boldsymbol{x}^\star$, score function $\phi$, number of mixture components $K$, number of Monte Carlo samples $L$;

1: $\boldsymbol{\alpha}, \boldsymbol{\theta}, \boldsymbol{\beta}^1, \cdots, \boldsymbol{\beta}^K \leftarrow$ Initialize model parameters;
2: **while** the likelihood is not converged **do**
3:     **for** $n = 1, 2, \cdots, N; i = 1, 2, \cdots, L$ **do**
4:         $\hat{\delta}_n \leftarrow |\boldsymbol{\sigma}_n|(|\boldsymbol{\sigma}_n| + 1)^{-2}$;
5:         $\boldsymbol{z}_n^{(i)} \leftarrow$ Generate a sample uniformly from $\mathcal{S}_{\boldsymbol{\sigma}_n}^{\hat{\delta}_n}$;
6:         $q(c_n^{(i)}) \leftarrow$ Infer the posterior of latent variable $c_n^{(i)}$ as in Eq. (9);
7:         $\boldsymbol{\alpha}, \boldsymbol{\theta}, \boldsymbol{\beta}^1, \cdots, \boldsymbol{\beta}^K \leftarrow$ Update the model parameters as in Eq. (10);
8: $\boldsymbol{d}^\star \leftarrow$ Predict the label distribution for instance $\boldsymbol{x}^\star$ as in Eq. (11);
9: **return** the label distribution $\boldsymbol{d}^\star$ for instance $\boldsymbol{x}^\star$;

---

### 3.1.3 Learning algorithm

We consider minimizing the cross-entropy of $p(\boldsymbol{d}|\boldsymbol{x})$ relative to $p^\star(\boldsymbol{d})$, i.e., maximizing $\mathbb{E}_{p^\star(\boldsymbol{d})}[\ln p(\boldsymbol{d}|\boldsymbol{x})]$. Since $p^\star(\boldsymbol{d})$ is usually a complex distribution and $\mathbb{E}_{p^\star(\boldsymbol{d})}[\ln p(\boldsymbol{d}|\boldsymbol{x})]$ involves the integration of $p^\star(\boldsymbol{d})$, it is often intractable. Therefore, we approximate it by the importance sampling method (whose detailed derivation can be found in the appendix):

$$\mathbb{E}_{p^\star(\boldsymbol{d})}[\ln p(\boldsymbol{d}|\boldsymbol{x})] \approx \sum_{i=1}^L \frac{\phi(\boldsymbol{d}^{(i)}; \boldsymbol{\theta}) \ln p(\boldsymbol{d}^{(i)}|\boldsymbol{x})}{\sum_{j=1}^L \phi(\boldsymbol{d}^{(j)}; \boldsymbol{\theta})}, \quad \boldsymbol{d}^{(i)} = \frac{1}{Z^{(i)}} \boldsymbol{z}^{(i)}, \quad \boldsymbol{z}^{(i)} \sim \text{Uni}(\boldsymbol{z}|\mathcal{S}_{\boldsymbol{\sigma}}^{\hat{\delta}}),$$

where $\boldsymbol{z}^{(i)} \sim \text{Uni}(\boldsymbol{z}|\mathcal{S}_{\boldsymbol{\sigma}}^{\hat{\delta}})$ denotes that $\boldsymbol{z}^{(i)}$ is sampled uniformly from $\mathcal{S}_{\boldsymbol{\sigma}}^{\hat{\delta}}$, and $Z^{(i)}$ equals to the sum of all elements in $\boldsymbol{z}^{(i)}$. Since our model contains discrete latent variables, we use the EM algorithm [8] to train the model. We introduce the variational distribution $q(c_n^{(i)})$ and obtain

$$\ln p(\boldsymbol{d}_n^{(i)}|\boldsymbol{x}_n) = \underbrace{\mathbb{E}_{q(c_n^{(i)})}\left[\ln \frac{p(\boldsymbol{d}_n^{(i)}|c_n^{(i)}, \boldsymbol{x}_n)p(c_n^{(i)}|\boldsymbol{x}_n)}{q(c_n^{(i)})}\right]}_{\text{ELBO (Evidence Lower Bound)}} + \underbrace{\text{KL}\left(q(c_n^{(i)})\|p(c_n^{(i)}|\boldsymbol{d}_n^{(i)}, \boldsymbol{x}_n)\right)}_{c\text{-posterior error}}.$$

We then alternate between M-step (maximizing ELBO) and E-step (minimizing the $c$-posterior error).

**E-step**: Infer the posterior of latent variables to minimize the $c$-posterior error:

$$\gamma_{nk}^{(i)} \triangleq q(c_n^{(i)} = k) = \frac{f_k(\boldsymbol{x}_n; \boldsymbol{\alpha})\text{Dir}(\boldsymbol{d}_n^{(i)}|f(\boldsymbol{x}_n; \boldsymbol{\beta}^k))}{\sum_{j=1}^K f_j(\boldsymbol{x}_n; \boldsymbol{\alpha})\text{Dir}(\boldsymbol{d}_n^{(i)}|f(\boldsymbol{x}_n; \boldsymbol{\beta}^j))}, \quad k \in [K]. \tag{9}$$

**M-step**: Update the model parameters to maximize the ELBO:

$$\underset{\boldsymbol{\alpha}, \boldsymbol{\theta}, \boldsymbol{\beta}^1, \cdots, \boldsymbol{\beta}^K}{\arg\max} \sum_{i=1}^L \sum_{n=1}^N \frac{\phi(\boldsymbol{d}_n^{(i)}; \boldsymbol{\theta})}{\sum_{j=1}^L \phi(\boldsymbol{d}_n^{(j)}; \boldsymbol{\theta})} \left(\sum_{k=1}^K \gamma_{nk}^{(i)} \ln \frac{f_k(\boldsymbol{x}_n; \boldsymbol{\alpha})\text{Dir}(\boldsymbol{d}_n^{(i)}|f(\boldsymbol{x}_n; \boldsymbol{\beta}^k))}{\gamma_{nk}^{(i)}}\right). \tag{10}$$

Once the model parameters are learned, we can predict the label distribution $\boldsymbol{d}$ for the test instance $\boldsymbol{x}^\star$ according to Eq. (8). To evaluate DRAM, we take the expectation of the label distribution based on $p(\boldsymbol{d}|\boldsymbol{x}^\star)$ as a deterministic output:

$$\boldsymbol{d}^\star = \sum_{k=1}^K \frac{1}{Z_k} f_k(\boldsymbol{x}; \boldsymbol{\alpha}) f(\boldsymbol{x}^\star, \boldsymbol{\beta}^k), \quad Z_k = \sum_{i=1}^M f_i(\boldsymbol{x}^\star, \boldsymbol{\beta}^k). \tag{11}$$

The overall learning process is shown in the Algorithm 1.

### 3.1.4 Instantiation: DRAM with linear learner and noninformative scoring function

Here we consider a concrete instantiation for our framework. For simplicity, we use the noninformative scoring function, i.e., $\phi(\boldsymbol{d}; \boldsymbol{\theta}) = 1$. $f(\boldsymbol{x}; \boldsymbol{\alpha})$ and $f(\boldsymbol{x}; \boldsymbol{\beta}^k)$ are modelled as linear models with

the softmax and softplus activation functions, respectively. To avoid over-fitting, we regularize the parameters $\{\boldsymbol{\beta}^k\}_{k=1}^K$ by $L_2$ norm. Then Eq. (10) can be rewritten as:

$$\underset{\boldsymbol{\alpha},\boldsymbol{\beta}^1,\cdots,\boldsymbol{\beta}^K}{\arg\min} \quad \lambda N \sum_{k=1}^K \left\|\boldsymbol{\beta}^k\right\|_2^2 - \frac{1}{L}\sum_{i=1}^L \sum_{n=1}^N \sum_{k=1}^K \gamma_{nk}^{(i)} \ln\left(f_k(\boldsymbol{x}_n;\boldsymbol{\alpha})\mathrm{Dir}(\boldsymbol{d}_n^{(i)}|f(\boldsymbol{x}_n;\boldsymbol{\beta}^k))\right),$$

$$f_k(\boldsymbol{x}_n;\boldsymbol{\alpha}) = \frac{\exp(\boldsymbol{\alpha}_k^\top \boldsymbol{x}_n')}{\sum_{j=1}^K \exp(\boldsymbol{\alpha}_j^\top \boldsymbol{x}_n')}, \quad f_j(\boldsymbol{x}_n;\boldsymbol{\beta}^k) = \ln(1+\exp(\boldsymbol{\beta}_j^{k\top}\boldsymbol{x}_n')), \quad \boldsymbol{x}_n' = [\boldsymbol{x}_n;1],$$

(12)

where $\boldsymbol{\alpha}_k$ is the $k$-th parameter vector in $\boldsymbol{\alpha}$, $\boldsymbol{\beta}_j^k$ is the $j$-th parameter vector in $\boldsymbol{\beta}^k$, and $\lambda$ is a trade-off parameter. We use L-BFGS [17] to optimize Eq. (12). We denote this instantiation as DRAM-LN.

### 3.2 Comparison method: data transformation

Since there is no existing method that can directly predict label distribution from multi-label ranking, we propose a comparison method called DT (dataset transformation). The main idea of DT is to transform multi-label rankings into logical labels. Then, we can recover label distributions by any existing LE algorithm and learn a label distribution predictor by any existing LDL algorithm. Specifically, the multi-label ranking dataset $\{(\boldsymbol{x}_n,\boldsymbol{\sigma}_n)\}_{n=1}^N$ will be transformed into $\bigcup_{n=1}^N \left\{\left(\boldsymbol{x}_n, \sum_{j\in\boldsymbol{\sigma}_{n,:i}}\boldsymbol{v}_j, |\boldsymbol{\sigma}_n|^{-1}\right)\right\}_{i=1}^{|\boldsymbol{\sigma}_n|}$, where $\boldsymbol{\sigma}_{n,:i}$ is the set of the 1st, 2nd, $\cdots$, and $i$-th elements in $\boldsymbol{\sigma}_n$ vector, $\boldsymbol{v}_j$ is an $|\boldsymbol{\sigma}_n|$-dimensional one-hot vector with the $j$-th element being 1, and the instance weight $|\boldsymbol{\sigma}_n|^{-1}$ is to avoid that the model learning favours instances with more relevant labels.

## 4 Related work

Our research is mainly related to LE [34] and LDL [4]. LE is a process of recovering label distributions from logical labels in the dataset. Most of the existing LE algorithms follow some basic assumptions. For example, some LE algorithms [9, 15, 22, 34] assume that instances with similar features have similar label distributions; some LE algorithms [12, 32] assume that semantically related labels also have close values in the label distribution; some LE algorithms [33, 16] assume that the label distribution is the low-dimensional representation of feature and logical label vectors.

LDL is the learning process on the instances annotated by label distributions. The LDL problem can be addressed in two ways. The first is to directly design algorithms that match the prerequisites of the LDL problem. Some prominent examples include the LDL algorithms [11, 19, 20, 25, 37] that mine label correlation and the LDL algorithms that maintain label ranking [13] or classification accuracy [26, 27, 28]. Another way is to extend existing learning algorithms. For example, LDSVR [5] fits each component of the label distribution by a support vector machine; LDLLogitBoost [31] extends the boosting method by additive weighted regressors; LDLF [21] designs a normalization layer to model the multi-modal distribution by extending differentiable decision trees.

## 5 Experiments

### 5.1 Datasets and evaluation measures

We adopt several widely used label distribution datasets, including Movie [4], Emotion6 [18], Twitter-LDL, and Flickr-LDL [36].[5] We manually reduce the label distributions in these datasets to multi-label rankings, train the model on these multi-label rankings, and then evaluate the model using the original label distributions in these datasets.[6] In addition, we create a dataset called NSRD (**N**atural **S**cene with multi-label **R**ankings and label **D**istributions). The instances and label sets in NSRD are the same as in the Natural-Scene [7]. Three experts are requested to annotate the instances with multi-label rankings and label distributions. Then we can directly train and evaluate the model using the multi-label rankings and label distributions, respectively. Details of these datasets can be found in the

---

[5]Although there are many label distribution datasets, we only adopt those whose label distributions are generated by expert annotation rather than algorithms or experimental instruments.

[6]These datasets contain some label distributions with identical values, e.g., $[0.3, 0.3, 0.4, 0]^\top$, which does not satisfy the prerequisites of this paper, so we remove them and their corresponding feature vectors beforehand.

Table 1: Experimental results ((rank) mean±std $t$-test) evaluated by four measures.

| Dataset | Method | Cheb ($\downarrow$) | Canber ($\downarrow$) | Cosine ($\uparrow$) | Rho ($\uparrow$) |
|---------|--------|---------------------|-----------------------|---------------------|------------------|
| Movie | DRAM-LN | (2) $0.124 \pm 0.001$ | (2) $1.058 \pm 0.008$ | (2) $0.932 \pm 0.001$ | (1) $0.720 \pm 0.006$ |
| | DT+VI+SA | (6) $0.163 \pm 0.001$ • | (6) $1.337 \pm 0.007$ • | (6) $0.888 \pm 0.002$ • | (5) $0.685 \pm 0.020$ • |
| | DT+VI+DM | (7) $0.166 \pm 0.003$ • | (7) $1.355 \pm 0.017$ • | (7) $0.884 \pm 0.004$ • | (4) $0.712 \pm 0.006$ • |
| | DT+GL+SA | (4) $0.143 \pm 0.001$ • | (5) $1.207 \pm 0.003$ • | (4) $0.912 \pm 0.001$ • | (6) $0.669 \pm 0.010$ • |
| | DT+GL+DM | (3) $0.142 \pm 0.001$ • | (4) $1.204 \pm 0.004$ • | (3) $0.915 \pm 0.001$ • | (2) $0.717 \pm 0.006$ • |
| | GT+SA | (5) $0.144 \pm 0.003$ • | (3) $1.201 \pm 0.021$ • | (5) $0.903 \pm 0.003$ • | (7) $0.625 \pm 0.010$ • |
| | GT+DM | (1) $0.114 \pm 0.001$ ○ | (1) $0.990 \pm 0.007$ ○ | (1) $0.936 \pm 0.001$ ○ | (3) $0.715 \pm 0.006$ • |
| Emotion6 | DRAM-LN | (1) $0.282 \pm 0.004$ | (1) $3.953 \pm 0.050$ | (1) $0.785 \pm 0.006$ | (1) $0.588 \pm 0.009$ |
| | DT+VI+SA | (2) $0.316 \pm 0.007$ • | (3) $3.984 \pm 0.073$ • | (3) $0.743 \pm 0.015$ • | (3) $0.506 \pm 0.035$ • |
| | DT+VI+DM | (3) $0.319 \pm 0.006$ • | (2) $3.966 \pm 0.065$ | (2) $0.748 \pm 0.012$ • | (2) $0.564 \pm 0.051$ |
| | DT+GL+SA | (5) $0.336 \pm 0.007$ • | (5) $4.121 \pm 0.041$ • | (5) $0.685 \pm 0.014$ • | (7) $0.319 \pm 0.041$ • |
| | DT+GL+DM | (4) $0.332 \pm 0.007$ • | (4) $4.047 \pm 0.037$ • | (4) $0.708 \pm 0.010$ • | (6) $0.386 \pm 0.035$ • |
| | GT+SA | (7) $0.567 \pm 0.011$ • | (7) $5.789 \pm 0.045$ • | (7) $0.494 \pm 0.016$ • | (5) $0.406 \pm 0.034$ • |
| | GT+DM | (6) $0.380 \pm 0.009$ • | (6) $4.769 \pm 0.043$ • | (6) $0.643 \pm 0.014$ • | (4) $0.473 \pm 0.029$ • |
| Twitter-LDL | DRAM-LN | (1) $0.355 \pm 0.009$ | (1) $6.526 \pm 0.018$ | (1) $0.828 \pm 0.009$ | (1) $0.604 \pm 0.005$ |
| | DT+VI+SA | (6) $0.546 \pm 0.003$ • | (4) $6.604 \pm 0.020$ • | (6) $0.621 \pm 0.005$ • | (7) $0.506 \pm 0.019$ • |
| | DT+VI+DM | (7) $0.580 \pm 0.010$ • | (6) $6.638 \pm 0.023$ • | (7) $0.549 \pm 0.027$ • | (4) $0.559 \pm 0.011$ • |
| | DT+GL+SA | (4) $0.520 \pm 0.003$ • | (2) $6.545 \pm 0.018$ • | (3) $0.682 \pm 0.001$ • | (3) $0.578 \pm 0.005$ • |
| | DT+GL+DM | (5) $0.534 \pm 0.003$ • | (3) $6.559 \pm 0.018$ • | (4) $0.656 \pm 0.001$ • | (2) $0.592 \pm 0.006$ • |
| | GT+SA | (3) $0.436 \pm 0.017$ • | (7) $6.937 \pm 0.026$ • | (5) $0.653 \pm 0.020$ • | (6) $0.516 \pm 0.009$ • |
| | GT+DM | (2) $0.372 \pm 0.004$ • | (5) $6.626 \pm 0.017$ • | (2) $0.763 \pm 0.006$ • | (5) $0.554 \pm 0.006$ • |
| Flickr-LDL | DRAM-LN | (1) $0.324 \pm 0.005$ | (1) $6.013 \pm 0.017$ | (1) $0.815 \pm 0.004$ | (1) $0.627 \pm 0.006$ |
| | DT+VI+SA | (6) $0.456 \pm 0.005$ • | (4) $6.116 \pm 0.021$ • | (5) $0.657 \pm 0.006$ • | (6) $0.542 \pm 0.021$ • |
| | DT+VI+DM | (7) $0.472 \pm 0.011$ • | (5) $6.146 \pm 0.040$ • | (7) $0.629 \pm 0.021$ • | (2) $0.592 \pm 0.010$ • |
| | DT+GL+SA | (4) $0.440 \pm 0.005$ • | (2) $6.076 \pm 0.021$ • | (3) $0.690 \pm 0.003$ • | (4) $0.573 \pm 0.005$ • |
| | DT+GL+DM | (5) $0.450 \pm 0.005$ • | (3) $6.090 \pm 0.020$ • | (4) $0.674 \pm 0.003$ • | (3) $0.592 \pm 0.006$ • |
| | GT+SA | (3) $0.439 \pm 0.008$ • | (7) $6.730 \pm 0.020$ • | (6) $0.634 \pm 0.010$ • | (7) $0.524 \pm 0.007$ • |
| | GT+DM | (2) $0.363 \pm 0.004$ • | (6) $6.360 \pm 0.014$ • | (2) $0.720 \pm 0.005$ • | (5) $0.554 \pm 0.005$ • |
| NSRD-e1 | DRAM-LN | (3) $0.509 \pm 0.006$ | (1) $7.649 \pm 0.017$ | (3) $0.599 \pm 0.009$ | (2) $0.459 \pm 0.013$ |
| | DT+VI+SA | (5) $0.576 \pm 0.008$ • | (7) $7.835 \pm 0.028$ • | (6) $0.462 \pm 0.013$ • | (7) $0.187 \pm 0.017$ • |
| | DT+VI+DM | (7) $0.595 \pm 0.007$ • | (6) $7.813 \pm 0.029$ • | (7) $0.459 \pm 0.008$ • | (6) $0.240 \pm 0.034$ • |
| | DT+GL+SA | (4) $0.574 \pm 0.006$ • | (4) $7.741 \pm 0.028$ • | (4) $0.523 \pm 0.004$ • | (4) $0.442 \pm 0.012$ • |
| | DT+GL+DM | (6) $0.579 \pm 0.006$ • | (5) $7.755 \pm 0.027$ • | (5) $0.509 \pm 0.007$ • | (1) $0.462 \pm 0.013$ |
| | GT+SA | (1) $0.468 \pm 0.008$ ○ | (3) $7.700 \pm 0.021$ • | (1) $0.610 \pm 0.012$ ○ | (3) $0.447 \pm 0.008$ • |
| | GT+DM | (2) $0.488 \pm 0.010$ ○ | (2) $7.659 \pm 0.035$ | (2) $0.604 \pm 0.014$ | (5) $0.438 \pm 0.013$ • |
| NSRD-e2 | DRAM-LN | (3) $0.509 \pm 0.006$ | (1) $7.649 \pm 0.017$ | (3) $0.599 \pm 0.009$ | (2) $0.459 \pm 0.013$ |
| | DT+VI+SA | (5) $0.570 \pm 0.006$ • | (7) $7.811 \pm 0.024$ • | (6) $0.469 \pm 0.018$ • | (7) $0.198 \pm 0.028$ • |
| | DT+VI+DM | (7) $0.588 \pm 0.009$ • | (6) $7.793 \pm 0.030$ • | (7) $0.465 \pm 0.012$ • | (6) $0.251 \pm 0.026$ • |
| | DT+GL+SA | (4) $0.568 \pm 0.005$ • | (4) $7.725 \pm 0.025$ • | (4) $0.525 \pm 0.004$ • | (4) $0.444 \pm 0.016$ • |
| | DT+GL+DM | (6) $0.574 \pm 0.006$ • | (5) $7.739 \pm 0.028$ • | (5) $0.511 \pm 0.008$ • | (1) $0.462 \pm 0.014$ |
| | GT+SA | (1) $0.461 \pm 0.011$ ○ | (3) $7.680 \pm 0.028$ • | (1) $0.617 \pm 0.018$ ○ | (3) $0.450 \pm 0.012$ • |
| | GT+DM | (2) $0.485 \pm 0.012$ ○ | (2) $7.664 \pm 0.035$ | (2) $0.605 \pm 0.014$ | (5) $0.438 \pm 0.014$ • |
| NSRD-e3 | DRAM-LN | (3) $0.554 \pm 0.011$ | (1) $7.699 \pm 0.023$ | (3) $0.577 \pm 0.013$ | (2) $0.455 \pm 0.012$ |
| | DT+VI+SA | (4) $0.615 \pm 0.009$ • | (7) $7.845 \pm 0.031$ • | (6) $0.456 \pm 0.018$ • | (7) $0.204 \pm 0.023$ • |
| | DT+VI+DM | (7) $0.638 \pm 0.005$ • | (6) $7.817 \pm 0.013$ • | (7) $0.446 \pm 0.013$ • | (6) $0.226 \pm 0.049$ • |
| | DT+GL+SA | (5) $0.619 \pm 0.004$ • | (4) $7.760 \pm 0.021$ • | (4) $0.504 \pm 0.004$ • | (5) $0.437 \pm 0.015$ • |
| | DT+GL+DM | (6) $0.624 \pm 0.005$ • | (5) $7.771 \pm 0.019$ • | (5) $0.493 \pm 0.010$ • | (1) $0.455 \pm 0.016$ |
| | GT+SA | (1) $0.490 \pm 0.012$ ○ | (3) $7.748 \pm 0.028$ • | (1) $0.601 \pm 0.019$ ○ | (3) $0.443 \pm 0.011$ • |
| | GT+DM | (2) $0.520 \pm 0.009$ ○ | (2) $7.701 \pm 0.030$ | (2) $0.599 \pm 0.016$ ○ | (4) $0.440 \pm 0.012$ • |

appendix. We used the six distance-based measures suggested in the paper [4] and a ranking-based measure suggested in the paper [13] to evaluate the performance of the model, which are Cheb (Chebyshev distance), Clark (Clark distance), Canber (Canberra distance), KL (Kullback-Leibler divergence), Cosine (cosine coefficient), Intersec (intersection similarity), and Rho (Spearman's rho coefficient). Due to page limitations, we only show the results on Cheb, Canber, Cosine, and Rho. Results on other measures are similar.

## 5.2 Comparison methods

On the one hand, we compare DRAM with the baseline method DT proposed in Section 3.2. GL (Graph Laplacian LE) [34] and SA (specialized LDL algorithm with BFGS optimizer) [4] are the classical LE and LDL algorithms respectively. VI (LE with variational inference) [33] and DM (LDL with label distribution manifold) [25] are the state-of-the-art LE and LDL algorithms, respectively. We combine them in pairs to construct four comparison methods, i.e., DT+GL+SA, DT+GL+DM, DT+VI+SA, and DT+VI+DM. The hyperparameter configuration of GL, VI and DM

Table 2: Average ranks of methods.

| Method | Cheb | Canber | Cosine | Rho |
|--------|------|--------|--------|-----|
| DRAM-LN | 2.00 | 1.14 | 2.00 | 1.43 |
| DT+GL+DM | 5.00 | 4.14 | 4.29 | 2.29 |
| DT+GL+SA | 4.29 | 3.71 | 3.86 | 4.71 |
| DT+VI+DM | 6.43 | 5.43 | 6.29 | 4.29 |
| DT+VI+SA | 4.86 | 5.43 | 5.43 | 6.00 |
| GT+DM | 2.43 | 3.43 | 2.43 | 4.43 |
| GT+SA | 3.00 | 4.71 | 3.71 | 4.86 |

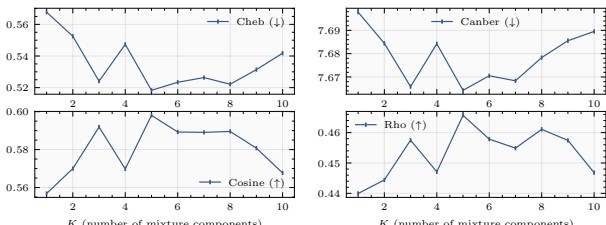

Figure 2: Performance with varying $K$ on NSRD.

follows their respective literature. For our method, we set $K = 3$ and $L = 20$, and $\lambda$ is selected from $\{10^{-5}, 5 \times 10^{-5}, 10^{-4}, 5 \times 10^{-4}, \cdots, 10^1, 5 \times 10^1\}$ by five-fold cross-validation. For the above comparison methods, since the label distributions are unavailable during training, the hyperparameter configuration that gives the highest Rho on the validation set will be used. On the other hand, we directly train DM and SA on the ground-truth label distributions for comparison. We refer to these two as GT+DM and GT+SA for short, respectively. For these two comparison methods, the hyperparameter configuration that gives the best Cheb, Canber, Cosine, and Rho on the validation set will be used. Each method is run for ten times on random dataset partitions (70% for training and 30% for test); the average values and standard derivations are recorded.

## 5.3 Results and discussions

Table 1 shows the four performance measures of each method on four reduced datasets and NSRD dataset. Since the NSRD is annotated by three experts, we can obtain three corresponding datasets, denoted by the "-e1", "-e2" and "-e3" suffixes, respectively; these three datasets have the same feature vectors and different annotation values. Each experimental result is formatted as "(rank) mean±std $t$-test"; "(rank)" denotes the rank of each method among the seven comparison methods; •/∘ indicates whether DRAM-LN is statistically superior/inferior to the corresponding methods (pairwise two-tailed $t$-test at 0.05 significance level); if neither • nor ∘ is shown, it means that there is no significant difference between the corresponding method and DRAM-LN; "↓" denotes "the lower the better", and "↑" denotes "the higher the better". Table 2 shows the average rank of each comparison method on each measure. Overall, our method achieves significant advantages. On the Emotion6, Twitter-LDL, and Flickr-LDL datasets, our method significantly outperforms almost any comparison methods. On the NSRD and Movie datasets, our method is only inferior to the GT-based methods on Cheb and Cosine measures. It is worth noting that DRAM-LN and DT-based methods outperform GT-based methods in many cases, such as the performance on Emotion6, Twitter-LDL, and Flickr-LDL datasets. We believe this is because some datasets are difficult to annotate; thus, the label distributions given by experts are noisy; then, fitting such label distributions exactly may lead to overfitting. This argument can be further supported by the fact that GT-based methods outperform our method on the NSRD dataset (where the label distributions are carefully annotated and less noisy).

Figure 2 shows how the number of mixture components $K$ affects the performance of our method on NSRD dataset. To save space, we show the average performance on NSRD-e1, NSRD-e2 and NSRD-e3 rather than showing them separately. It is obvious that the mixture model ($K > 1$) always outperform the single model ($K = 1$). In addition, it can be seen that appropriately increasing the Dirichlet components in the mixture can improve the model capacity and thus improve the predictive performance, but too many Dirichlet components may lead to overfitting and thus degrade the predictive performance.

## 6 Limitations and conclusion

**Limitations.** 1) EAE is defined for the label description vector and does not directly reflect the approximation error to the label distribution; this limitation arises because normalizing the label description vector to a label distribution will lead to an extremely complex closed form of EAE. We do not believe that this limitation have a significant impact on our main results since the approximation error to the label distribution does not exceed that to the label description vector. For example, if the true and estimated label description vectors are $z$ and $tz$ (where $t$ is a scaler), respectively, they

will produce non-zero errors w.r.t. the label description vector; but $z$ and $tz$ are the same after normalization, i.e., they do not produce errors w.r.t. the label distribution. 2) If several labels actually describe the instance to the same degree, then requiring experts to give the strict order of these labels may lead to errors and invalidate Theorem 1; we plan to extend the theoretical results to this case in the future. Fortunately, DRAM framework can suit this case by a minor modification on the explicit range, i.e., allowing the description degree of these labels to be identical in line 5 of Algorithm 1.

**Conclusion.** We derive some theorems and corollaries to reveal the relation between multi-label ranking and label distribution, and propose a generic framework, DRAM, for predicting label distribution from multi-label ranking. DRAM is cost-effective: It is trained on the examples with multi-label rankings and achieves performance comparable to that of LDL methods which require expensive label distribution annotations; DRAM is flexible: It allows users to easily encode their prior knowledge by a scoring function; DRAM is end-to-end: It integrates the processes of recovering and learning label distributions into one learning criterion, rather than performing them separately. Experimental results show the superiority of our proposal.

# 7    Acknowledgments

This work was partially supported by the National Key Research and Development Program of China under Grant 2019YFB1706900, the National Natural Science Foundation of China (62176123, U20B2064), and the Fundamental Research Funds for the Central Universities (30920021131).

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
