# Predicting Label Distribution from Multi-label Ranking

**Yunan Lu, Xiuyi Jia**
School of Computer Science and Engineering
Nanjing University of Science and Technology, Nanjing 210094, China
{luyn, jiaxy}@njust.edu.cn

## A   Appendix

### A.1   Proof of Theorem 1

**Theorem 1** *If an instance is annotated by a multi-label ranking $\boldsymbol{\sigma}$, $m$ is the number of relevant labels, $\delta$ and $\hat{\delta}$ are the implicit and explicit margins, respectively, then the EAE of $\boldsymbol{\sigma}$ is*

$$\epsilon_{\boldsymbol{\sigma}}^{\delta,\hat{\delta}} = \frac{m}{6(m+1)}\Big((m+1)^2(\delta^2+\hat{\delta}^2) - 2m(\delta+\hat{\delta}) - (4m+2)\delta\hat{\delta} + 2\Big). \tag{1}$$

*Proof.* The expected approximation error arising from multi-label ranking comes mainly from the relevant labels, hence we only need to consider the relevant labels. We denote the number of relevant labels as $m$, the true importance degree of label $y_{\boldsymbol{\sigma}_i}$ as $s_i$, and the estimated importance degree of label $y_{\boldsymbol{\sigma}_i}$ as $\hat{s}_i$. In order to ensure that $\forall i \in [m], s_i \in [\delta, 1]$, we must have $\forall i \in [m-1], s_i \in [i\delta, s_{i+1}-\delta]$ and $s_m \in [m\delta, 1]$. Similarly, $\forall i \in [m-1], \hat{s}_i \in [i\hat{\delta}, \hat{s}_{i+1}-\hat{\delta}]$ and $\hat{s}_m \in [m\hat{\delta}, 1]$. Therefore, we can obtain the volume of theh space $\mathcal{S}_{\boldsymbol{\sigma}}^{\hat{\delta}}$:

$$V_{\boldsymbol{\sigma}}^{\hat{\delta}} = \int_{m\hat{\delta}}^1 \int_{(m-1)\hat{\delta}}^{s_m-\hat{\delta}} \cdots \int_{\hat{\delta}}^{s_2-\hat{\delta}} \mathrm{d}s_1 \cdots \mathrm{d}s_{m-1}\mathrm{d}s_m. \tag{2}$$

We use the mathematical induction method to calculate $V_{\boldsymbol{\sigma}}^{\hat{\delta}}$. By observing the calculation results for the cases $m = 1, 2, 3, 4$, we make the following conjecture:

$$F_1(k) = \int_{(k-1)\hat{\delta}}^{s_k-\delta} \int_{(k-2)\hat{\delta}}^{s_{k-1}-\hat{\delta}} \cdots \int_{\hat{\delta}}^{s_2-\hat{\delta}} \mathrm{d}s_1 \cdots \mathrm{d}s_{k-2}\mathrm{d}s_{k-1} = \frac{(s_k - k\hat{\delta})^{k-1}}{(k-1)!}. \tag{3}$$

It is obvious that Eq. (3) holds for $k = 2$. For $k + 1$, we have:

$$F_1(k+1) = \int_{k\hat{\delta}}^{s_{k+1}-\hat{\delta}} \frac{(s_k - k\hat{\delta})^{k-1}}{(k-1)!}\mathrm{d}s_k = \frac{(s_{k+1} - (k+1)\hat{\delta})^k}{k!}. \tag{4}$$

Therefore, Eq. (3) holds for $k = 2, 3, \cdots$. Then we can obtain $V_{\boldsymbol{\sigma}}^{\hat{\delta}} = \frac{(1-m\hat{\delta})^m}{m!}$ by substituting $s_{k+1}$ in Eq. (4) for $1 + \hat{\delta}$. Similarly, we have $V_{\boldsymbol{\sigma}}^{\delta} = \frac{(1-m\delta)^m}{m!}$. Next we use the same idea to integrate the squared Euclidean distance between $[s_i]_{i=1}^m$ and $[\hat{s}_i]_{i=1}^m$. By observing the calculation results for the cases $m = 1, 2, 3, 4$, we make the following conjecture:

$$F_2(k) = \int_{(k-1)\delta}^{s_k-\delta} \cdots \int_{\delta}^{s_2-\delta} \int_{(k-1)\hat{\delta}}^{s_k-\hat{\delta}} \cdots \int_{\hat{\delta}}^{s_2-\hat{\delta}} \sum_{i=1}^{k-1}(s_i - \hat{s}_i)^2 \mathrm{d}\hat{s}_1\mathrm{d}\hat{s}_{k-1}\mathrm{d}s_1\mathrm{d}s_{k-1}$$

$$= \frac{\Big((k\delta - s_k)(k\hat{\delta} - s_k)\Big)^{k-1}}{6k!(k-2)!}\Big(k^2(\delta^2+\hat{\delta}^2) + 2k(s_k^2 + \hat{s}_k^2 - \delta s_k - \hat{\delta}\hat{s}_k) - (4k-2)\hat{s}_k s_k\Big). \tag{5}$$

It is obvious that Eq. (5) holds for $k = 2$. For $k + 1$, we have:

$$
\begin{aligned}
F_2(k+1) &= \int_{k\delta}^{s_{k+1}-\delta} \int_{k\hat{\delta}}^{\hat{s}_{k+1}-\hat{\delta}} F_2(k)\mathrm{d}s_k\mathrm{d}\hat{s}_k \\
&= \frac{\Big(((k+1)\delta - s_{k+1})((k+1)\hat{\delta} - s_{k+1})\Big)^k}{6(k+1)!(k-1)!} \cdot \Big((k+1)^2(\delta^2 + \hat{\delta}^2) \\
&\quad + 2(k+1)(s_{k+1}^2 + \hat{s}_{k+1}^2 - \delta s_{k+1} - \hat{\delta}\hat{s}_{k+1}) - (4k+2)\hat{s}_{k+1}s_{k+1}\Big).
\end{aligned}
\tag{6}
$$

Therefore, Eq. (5) holds for $k = 2, 3, \cdots$. By substituting $s_{k+1}$ and $\hat{s}_{k+1}$ for $1 + \delta$ and $1 + \hat{\delta}$, respectively, we have:

$$
\begin{aligned}
\int_{\boldsymbol{z}\in\mathcal{S}_{\boldsymbol{\sigma}}^{\delta}} \int_{\hat{\boldsymbol{z}}\in\mathcal{S}_{\boldsymbol{\sigma}}^{\hat{\delta}}} \|\boldsymbol{z} - \hat{\boldsymbol{z}}\|_2^2 \mathrm{d}\hat{\boldsymbol{z}}\mathrm{d}\boldsymbol{z} &= \frac{\Big((1-m\delta)(1-m\hat{\delta})\Big)^m}{6(m+1)!(m-1)!} \cdot \Big((m+1)^2(\delta^2 + \hat{\delta}^2) \\
&\quad - 2m(\delta + \hat{\delta}) - (4m+2)\delta\hat{\delta} + 2\Big).
\end{aligned}
\tag{7}
$$

Therefore, Eq. (1) can be obtained by combining Eq (7) and $V_{\boldsymbol{\sigma}}^{\hat{\delta}} V_{\boldsymbol{\sigma}}^{\delta} = \frac{(1-m\hat{\delta})^m(1-m\delta)^m}{(m!)^2}$. $\qquad\square$

## A.2 Proof of Lemma 1

**Lemma 1** *If an instance is annotated by a multi-label ranking $\boldsymbol{\sigma}$, then the margins $\delta$ and $\hat{\delta}$ satisfy that $0 \le \delta \le m^{-1}$ and $0 \le \hat{\delta} \le m^{-1}$.*

*Proof.* To ensure $\mathcal{S}_{\boldsymbol{\sigma}}^{\delta} \ne \varnothing$, we can obtain that there is at least one label importance vector $\boldsymbol{z}$ satisfying $(\forall k \in \boldsymbol{\sigma}, z_k \in [\delta, 1]) \wedge (\forall i \in [|\boldsymbol{\sigma}| - 1], z_{\sigma_i} \le z_{\boldsymbol{\sigma}_{i+1}} - \delta) \wedge (\forall j \in [M]\backslash\boldsymbol{\sigma}, z_j = 0)$. Accordingly, we can obtain

$$
\delta \le z_{\sigma_1} \le z_{\sigma_2} - \delta \le z_{\sigma_3} - 2\delta \le \cdots \le z_{\sigma_m} - (m-1)\delta \le 1 - (m-1)\delta.
$$

Therefore, $\delta \le 1 - (m-1)\delta$, i.e., $\delta \le \frac{1}{m}$. Similarly, $\hat{\delta} \le \frac{1}{m}$. $\qquad\square$

## A.3 Proof of Corollary 1

**Corollary 1** *If an instance is annotated by a multi-label ranking $\boldsymbol{\sigma}$, $m$ is the number of relevant labels, the explicit margin $\hat{\delta}^{\star}$ minimizing the EAE of $\boldsymbol{\sigma}$ is $\hat{\delta}^{\star} = ((2m+1)\delta + m)(m+1)^{-2}$.*

*Proof.* It is obvious that $\epsilon_{\boldsymbol{\sigma}}^{\delta,\hat{\delta}}$ is a quadratic function of $\hat{\delta}$ and the second order derivative of $\epsilon_{\boldsymbol{\sigma}}^{\delta,\hat{\delta}}$ w.r.t. $\hat{\delta}$ is a positive number, hence the only stationary point $\hat{\delta}$ of $\epsilon_{\boldsymbol{\sigma}}^{\delta,\hat{\delta}}$, i.e., $\hat{\delta}^{\star} = \frac{(2m+1)\delta+m}{(m+1)^2}$, is the optimal one that minimizes the expected approximation error. $\qquad\square$

## A.4 Proof of Corollary 2

**Corollary 2** *If an instance is annotated by a multi-label ranking $\boldsymbol{\sigma}$, $m$ is the number of relevant labels, $0 \le \delta \le m^{-1}$, $m(m+1)^{-2} \le \hat{\delta} \le m^{-1}$, then the EAE of $\boldsymbol{\sigma}$ is bounded by:*

$$
0 \le \epsilon_{\boldsymbol{\sigma}}^{\delta,\hat{\delta}} \le \frac{m(m^2+4m+2)}{6(m+1)^3} < \frac{1}{5}.
\tag{8}
$$

*Proof.* It is obvious that $\epsilon_{\boldsymbol{\sigma}}^{\delta,\hat{\delta}} \ge 0$ holds, and $\lim_{\delta\to\frac{1}{m},\hat{\delta}\to\frac{1}{m}} = 0$. Since $\epsilon_{\boldsymbol{\sigma}}^{\delta,\hat{\delta}} \ge 0$ is a quadratic function of $\hat{\delta}$ and $\delta$, and the second order derivative $\partial\epsilon_{\boldsymbol{\sigma}}^{\delta,\hat{\delta}}/\partial\hat{\delta} > 0$ and $\partial\epsilon_{\boldsymbol{\sigma}}^{\delta,\hat{\delta}}/\partial\delta > 0$, the maximum value of $\epsilon_{\boldsymbol{\sigma}}^{\delta,\hat{\delta}}$ is taken at the boundary of $\delta$ and $\hat{\delta}$. Therefore, we only need to check the following four equations, the largest of which is the maximum value of $\epsilon_{\boldsymbol{\sigma}}^{\delta,\hat{\delta}}$:

$$
\begin{aligned}
&\epsilon_{\boldsymbol{\sigma}}^{\delta,\hat{\delta}}\Big|_{\delta=0,\hat{\delta}=\frac{m}{(m+1)^2}} = \frac{m(m^2+4m+2)}{6(m+1)^3}, &&\epsilon_{\boldsymbol{\sigma}}^{\delta,\hat{\delta}}\Big|_{\delta=\frac{1}{m},\hat{\delta}=\frac{1}{m}} = 0, \\
&\epsilon_{\boldsymbol{\sigma}}^{\delta,\hat{\delta}}\Big|_{\delta=\frac{1}{m},\hat{\delta}=\frac{m}{(m+1)^2}} = \frac{(2m+1)^2}{6m(m+1)^3}, &&\epsilon_{\boldsymbol{\sigma}}^{\delta,\hat{\delta}}\Big|_{\delta=0,\hat{\delta}=\frac{1}{m}} = \frac{m+1}{6m}.
\end{aligned}
\tag{9}
$$

Obviously, $\epsilon_{\boldsymbol{\sigma}}^{\delta,\hat{\delta}}$ takes the maximum value when $\delta = 0$ and $\hat{\delta} = \frac{m}{(m+1)^2}$, i.e., $\epsilon_{\boldsymbol{\sigma}}^{\delta,\hat{\delta}} \le \frac{m(m^2+4m+2)}{6(m+1)^3}$. Further, it is easy to verify that the following formula holds for any positive integer $m$:

$$5m^3 + 20m^2 + 10m < 6m^3 + 18m^2 + 18m + 6, \tag{10}$$

then we have $\frac{m(m^2+4m+2)}{6(m+1)^3} < \frac{1}{5}$. Therefore, the formula (8) is proved. $\qquad\square$

### A.5 Proof of Theorem 2

**Theorem 2** *If an instance is annotated by a logical label vector $\boldsymbol{l}$, $m$ is the number of relevant labels, $\delta$ and $\hat{\delta}$ are the implicit and explicit margins, respectively, then the EAE of $\boldsymbol{l}$ is*

$$\epsilon_{\boldsymbol{l}}^{\delta,\hat{\delta}} = \frac{m}{6}(2\delta^2 + 2\hat{\delta}^2 - \delta - \hat{\delta} - 3\delta\hat{\delta} + 1). \tag{11}$$

*Proof.* The expected approximation error arising from logical labels comes mainly from labels with a logical value of $1$, hence we consider only the relevant labels. We denote the number of relevant labels as $m$, i.e., $m = \sum_{i=1}^{M} \mathbb{I}(l_i = 1)$. We first calculate $V_{\boldsymbol{l}}^{\hat{\delta}}$:

$$V_{\boldsymbol{l}}^{\hat{\delta}} = \int_{\hat{\delta}}^{1} \int_{\hat{\delta}}^{1} \cdots \int_{\hat{\delta}}^{1} \mathrm{d}z_1 \mathrm{d}z_2 \cdots \mathrm{d}z_m = (1 - \hat{\delta})^m. \tag{12}$$

In the same way, we can obtain $V_{\boldsymbol{l}}^{\delta} = (1 - \delta)^m$. In the following we integrate the squared Euclidean distance between $\boldsymbol{z}$ and $\hat{\boldsymbol{z}}$:

$$\int_{\boldsymbol{z} \in \mathcal{S}_{\boldsymbol{l}}^{\delta}} \int_{\hat{\boldsymbol{z}} \in \mathcal{S}_{\boldsymbol{l}}^{\hat{\delta}}} \sum_{i=1}^{m} (z_i - \hat{z}_i)^2 \mathrm{d}\hat{\boldsymbol{z}} \mathrm{d}\boldsymbol{z} = \int_{\delta}^{1} \cdots \int_{\delta}^{1} \int_{\hat{\delta}}^{1} \cdots \int_{\hat{\delta}}^{1} \sum_{i=1}^{m} (z_i - \hat{z}_i)^2 \mathrm{d}z_1 \cdots \mathrm{d}z_m \mathrm{d}\hat{z}_1 \cdots \mathrm{d}\hat{z}_m$$

$$= \frac{m}{6}(1 - \delta)^m (1 - \hat{\delta})^m (2\delta^2 + 2\hat{\delta}^2 - \delta - \hat{\delta} - 3\delta\hat{\delta} + 1). \tag{13}$$

Finally, we can obtain the EAE of $\boldsymbol{l}$ by combining Eq. (12) and Eq. (13). $\qquad\square$

### A.6 Proof of Corollary 3

**Corollary 3** *If an instance is annotated by a multi-label ranking $\boldsymbol{\sigma}$, $m$ is the number of relevant labels, $\delta$ and $\hat{\delta}$ are uniform over $\left[0, m^{-1}\right]$ and $\left[m(m+1)^{-2}, m^{-1}\right]$, respectively, then we have:*

$$\underset{\delta,\hat{\delta}}{\mathbb{E}}\left[\epsilon_{\boldsymbol{\sigma}}^{\delta,\hat{\delta}}\right] = \frac{2m^4 + 8m^3 + 8m^2 + 4m + 1}{36m(m+1)^3}. \tag{14}$$

*Proof.*

$$\underset{\delta,\hat{\delta}}{\mathbb{E}}\left[\epsilon_{\boldsymbol{\sigma}}^{\delta,\hat{\delta}}\right] = m\left(\frac{1}{m} - \frac{m}{(m+1)^2}\right)^{-1} \int_{0}^{\frac{1}{m}} \int_{\frac{m}{(m+1)^2}}^{\frac{1}{m}} \text{Eq. (1)} \mathrm{d}\hat{\delta}\mathrm{d}\delta$$

$$= \frac{2m^4 + 8m^3 + 8m^2 + 4m + 1}{36m(m+1)^3}. \tag{15}$$

$\square$

### A.7 Proof of Corollary 4

**Corollary 4** *Suppose that $\epsilon_{\boldsymbol{l}}^{\delta,\hat{\delta}_l}$ and $\epsilon_{\boldsymbol{\sigma}}^{\delta,\hat{\delta}_\sigma}$ are the EAE of the logical label vector $\boldsymbol{l}$ and the EAE of the multi-label ranking $\boldsymbol{\sigma}$, respectively, we have the following inequality holds for $m \ge 3$:*

$$\epsilon_{\boldsymbol{l}}^{\delta,\hat{\delta}_l} - \epsilon_{\boldsymbol{\sigma}}^{\delta,\hat{\delta}_\sigma} \ge \frac{7m}{48}(\delta^2 - 2\delta) + \frac{m(m-1)(7m^2 + 20m + 9)}{48(m+1)^3}$$

$$> \frac{7m^5 - m^4 - 46m^3 - 30m^2 + 7m + 7}{48m(m+1)^3} > 0. \tag{16}$$

*Proof.* Since $\epsilon_l^{\delta,\hat{\delta}_l}$ is a quadratic function of $\hat{\delta}_l$ and the coefficient of the quadratic term is a positive number, the minimum value of $\epsilon_l^{\delta,\hat{\delta}_l}$ w.r.t. $\hat{\delta}_l$ is $\frac{7m(\delta-1)^2}{48}$. According to Corollary 2, we have $\epsilon_{\boldsymbol{\sigma}}^{\delta,\hat{\delta}_{\boldsymbol{\sigma}}} \leq \frac{m(m^2+4m+2)}{6(m+1)^3}$. Then we have

$$\epsilon_l^{\delta,\hat{\delta}_l} - \epsilon_{\boldsymbol{\sigma}}^{\delta,\hat{\delta}_{\boldsymbol{\sigma}}} \geq \frac{7m}{48}(\delta^2 - 2\delta) + \frac{m(m-1)(7m^2+20m+9)}{48(m+1)^3}. \tag{17}$$

Since $\delta^2 - 2\delta > \frac{1}{m^2} - \frac{2}{m}$, we can obtain the Eq (16) by substituting $\delta$ for $\frac{1}{m}$. Obviously, $\frac{7m^5-m^4-46m^3-30m^2+7m+7}{48m(m+1)^3} > 0$ holds for $m \geq 3$. □

## A.8 Details of DRAM

The probability density function of Dirichlet distribution is

$$\text{Dir}(\boldsymbol{d}|\boldsymbol{\mu}) = \frac{1}{B(\boldsymbol{\mu})}\prod_{i=1}^{M} d_i^{\mu_i-1}, \quad B(\boldsymbol{\mu}) = \frac{1}{\Gamma(\sum_{i=1}^{M}\mu_i)}\prod_{i=1}^{M}\Gamma(\mu_i), \quad \Gamma(\mu) = \int_0^{\infty} x^{\mu-1}e^{-x}\mathrm{d}x.$$

The mean of Dirichlet distribution is

$$\mathop{\mathbb{E}}_{\boldsymbol{d}\sim\text{Dir}(\boldsymbol{d}|\boldsymbol{\mu})}[\boldsymbol{d}] = \frac{1}{Z_{\boldsymbol{\mu}}}\boldsymbol{\mu}, \quad Z_{\boldsymbol{\mu}} = \sum_{i=1}^{M}\mu_i.$$

## A.9 Monte Carlo Approximation for $\mathbb{E}_{p^\star(\boldsymbol{d})}\left[\ln p(\boldsymbol{d}|\boldsymbol{x})\right]$

Let the importance sampling distribution be $\tilde{p}(\boldsymbol{d}) = \frac{1}{Z_{\tilde{p}}}\int_0^{\infty}\mathbb{I}(t\boldsymbol{d} \in \mathcal{S}_{\boldsymbol{\sigma}}^{\hat{\delta}})\mathrm{d}t$; then, the negative cross-entropy can be approximated by:

$$p^\star(\boldsymbol{d})/\tilde{p}(\boldsymbol{d}) = \frac{\frac{1}{Z_{p^\star}}\phi(\boldsymbol{d};\boldsymbol{\theta})\int_0^{\infty}\mathbb{I}(t\boldsymbol{d} \in \mathcal{S}_{\boldsymbol{\sigma}}^{\hat{\delta}})\mathrm{d}t}{\frac{1}{Z_{\tilde{p}}}\int_0^{\infty}\mathbb{I}(t\boldsymbol{d} \in \mathcal{S}_{\boldsymbol{\sigma}}^{\hat{\delta}})\mathrm{d}t} = \frac{Z_{\tilde{p}}}{Z_{p^\star}}\phi(\boldsymbol{d};\boldsymbol{\theta}),$$

$$\mathop{\mathbb{E}}_{p^\star(\boldsymbol{d})}\left[\ln p(\boldsymbol{d}|\boldsymbol{x})\right] \approx \sum_{i=1}^{L}\frac{\phi(\boldsymbol{d}^{(i)};\boldsymbol{\theta})}{\sum_{j=1}^{L}\phi(\boldsymbol{d}^{(j)};\boldsymbol{\theta})}\ln p(\boldsymbol{d}^{(i)}|\boldsymbol{x}). \tag{18}$$

We can draw samples from $\tilde{p}(\boldsymbol{d}) = \frac{1}{Z_{\tilde{p}}}\int_0^{\infty}\mathbb{I}(t\boldsymbol{d} \in \mathcal{S}_{\boldsymbol{\sigma}}^{\hat{\delta}})\mathrm{d}t$ as follows:

$$\boldsymbol{z}^{(i)} \sim \text{Uni}(\boldsymbol{z}|\mathcal{S}_{\boldsymbol{\sigma}}^{\hat{\delta}}), \quad \boldsymbol{d}^{(i)} = \frac{1}{Z^{(i)}}\boldsymbol{z}^{(i)}. \tag{19}$$

## A.10 Details of Experiments

The information of the datasets we used is shown in Table 1. The first four rows in Table 1 are the existing label distribution datasets; the last three rows in Table 1 are the datasets we created. Since some examples in the original label distribution datasets do not satisfy the prerequisites of our paper (i.e., there are some examples $(\boldsymbol{x}, \boldsymbol{d})$ such that there exist relevant labels with identical label description degrees), we remove these examples from the dataset to obtain such a dataset: $\{(\boldsymbol{x}, \boldsymbol{d}) \in \mathcal{D}|\forall(d_i \neq 0, d_j \neq 0), d_i \neq d_j\}$, where $\mathcal{D} = \{(\boldsymbol{x}_n, \boldsymbol{d}_n)\}_{n=1}^{N}$. In Table 1, $N_1 \rightarrow N_2$ means that the original dataset with $N_1$ instances is reduced to the dataset with $N_2$ instances. Since the instances in Emotion6, Twitter-LDL and Flickr-LDL are images, we use a VGG16 [2] network pre-trained on ImageNet [1] to extract 1000-dimensional features. For the NSRD dataset, we use the feature vectors suggested in [3]. Besides, we use the random search method as the hyperparameter optimization technique, and the number of searches is set to 30.

## References

[1] Olga Russakovsky, Jia Deng, Hao Su, Jonathan Krause, Sanjeev Satheesh, Sean Ma, Zhiheng Huang, Andrej Karpathy, Aditya Khosla, Michael S. Bernstein, Alexander C. Berg, and Li Fei-Fei. Imagenet large scale visual recognition challenge. *International Journal of Computer Vision*, 115:211–252, 2015.

Table 1: Statistics of datasets.

| Dataset | # Instances | # Features | # Labels |
|---------|-------------|------------|----------|
| Movie | $7755 \rightarrow 6437$ | 1869 | 5 |
| Emotion6 | $1980 \rightarrow 1063$ | 1000 | 7 |
| Twitter-LDL | $10045 \rightarrow 6147$ | 1000 | 8 |
| Flickr-LDL | $11150 \rightarrow 4212$ | 1000 | 8 |
| NSRD-e1 | 2000 | 135 | 9 |
| NSRD-e2 | 2000 | 135 | 9 |
| NSRD-e3 | 2000 | 135 | 9 |

[2] Karen Simonyan and Andrew Zisserman. Very deep convolutional networks for large-scale image recognition. In *International Conference on Learning Representations*, 2015.

[3] Zhi-Hua Zhou and Min-Ling Zhang. Multi-instance multi-label learning with application to scene classification. In *Advances in Neural Information Processing Systems*, pages 1609–1616, 2006.