# OpenReview forum: "Predicting Label Distribution from Multi-label Ranking"
_NeurIPS.cc/2022/Conference — NeurIPS 2022 Accept_

### Official Review · Reviewer_b3Gr · 2022-06-27

**Rating:** 7
**Confidence:** 3
**Soundness:** 3 good
**Presentation:** 3 good
**Contribution:** 3 good

**Summary:**

This paper aims at predicting label distribution from multi-label ranking. It is different from the existing label distribution learning (LDL) and label enhancement (LE), where LDL learns a predictive mapping on training set with label distributions annotations and LE generates label distributions from instances with logical labels. This work is a compromise w.r.t. annotation cost but has good guarantees for performance.

**Questions:**

I have listed some weaknesses above. Here, my further concern is why the general LDL data sets are not used in experiments. It is very easy to construct data sets with label rankings and with binary labels from LDL data set. The 15 LDL data sets release by Geng can be found at: http://palm.seu.edu.cn/xgeng/LDL/index.htm

**Limitations:**

Authors points out two limitations in Section 6. For the 2nd one, i.e., "If several labels actually describe the instance to the same degree, then requiring experts to give the strict order of these labels may lead to errors and invalidate Theorem 1", authors should extend the proposed framework to deal with such kind of data sets, not treat it as a future work. After all, it is more realistic that some labels have the same description degree.

**Strengths And Weaknesses:**

Strengths:
1. This paper has a good motivation.
2. This paper proposed a framework of label distribution predicting from multi-label ranking.
3. Some experimental results validate the superiority of the proposed method.

Weaknesses:
1. It seems that the theoretical analysis has little relatedness with the proposed framework and there are not experiments to support the theoretical results. It is something that looks beautiful.
2. The trade-off parameter $\lambda$ is selected from a very wide range by five-fold cross-validation. Is there a good default choice? Parameter sensitivity should be made.  Besides, five-fold cross-validation is conducted over the whole data set or only the 70% data for training?

---

> ### Author Response · Authors · 2022-07-29
> **Response to Reviewer b3Gr**
>
> We appreciate your suggestions. Below we give the point-by-point responses to your questions.
>
> **Q1: It seems that the theoretical analysis has little relatedness with the proposed framework and there are not experiments to support the theoretical results. It is something that looks beautiful.**
>
> A: The theoretical part of our paper serves two purposes: (1) It helps us understand the advantages of multi-label ranking w.r.t. approximating the true label distribution. (2) The optimal range of $\\hat\\delta$ can help us construct the high-quality prior distribution of the label distribution, i.e., Eq. (7) in our paper.
>
> **Q2: The trade-off parameter $\\lambda$ is selected from a very wide range by five-fold cross-validation. Is there a good default choice? Parameter sensitivity should be made. Besides, five-fold cross-validation is conducted over the whole data set or only the 70% data for training?**
>
> A: (1) The parameter sensitivity is shown in the following table, where each entry denotes the performances on Cheb/Rho. The last column NSRD is the average performance on NSRD-e1, NSRD-e2 and NSRD-e3 datasets. It can be seen that $\\lambda=1$ is a good default choice for most datasets. (2) Five-fold cross-validation is conducted over the 70% data for training. Specifically, we first select the optimal hyperparameter on the training set (70% dataset) using five-fold cross-validation, then train the model on the training set using that hyperparameter, and finally report the performance on the test set (30% dataset).
>
> |$\\lambda$|Emotion6| Flickr-LDL|Twitter-LDL|Movie| NSRD|
> |-|-|-|-|-|-|
> |1.0E-05|0.304/0.486|0.352/0.488|0.372/0.454|0.124/0.720|0.539/0.154|
> |5.0E-05|0.304/0.484|0.353/0.488|0.355/0.481|0.124/0.719|0.543/0.150|
> |1.0E-04|0.304/0.486|0.353/0.487|0.347/0.493|0.124/0.719|0.549/0.152|
> |5.0E-04|0.304/0.484|0.352/0.488|0.322/0.524|0.125/0.717|0.553/0.150|
> |1.0E-03|0.304/0.484|0.352/0.489|0.313/0.538|0.126/0.716|0.555/0.084|
> |5.0E-03|0.301/0.494|0.351/0.492|0.303/0.556|0.129/0.712|0.549/0.202|
> |1.0E-02|0.298/0.502|0.346/0.501|0.303/0.563|0.130/0.707|0.556/0.295|
> |5.0E-02|0.284/0.542|0.323/0.546|0.307/0.575|0.141/0.651|0.580/0.259|
> |1.0E-01|0.281/0.564|0.312/0.588|0.310/0.579|0.147/0.649|0.570/0.311|
> |5.0E-01|0.282/0.588|0.318/0.620|0.319/0.589|0.164/0.648|0.554/0.418|
> |1.0E+00|0.283/0.581|0.324/0.627|0.326/0.591|0.171/0.648|0.553/0.426|
> |5.0E+00|0.299/0.571|0.352/0.620|0.343/0.599|0.179/0.648|0.550/0.438|
> |1.0E+01|0.309/0.555|0.364/0.615|0.355/0.604|0.181/0.648|0.546/0.439|
> |5.0E+01|0.328/0.527|0.396/0.603 | 0.366/0.601|0.182/0.648|0.524/0.453|
>
> **Q3: Why the general LDL data sets are not used in experiments. It is very easy to construct data sets with label rankings and with binary labels from LDL data set. The 15 LDL data sets release by Geng**
>
> A: Our criterion for selecting the datasets is that the label distributions of the datasets are generated by expert annotation, which is in line with the motivation of our paper. In terms of the 15 datasets published by Geng, the label distributions of Natural-Scene dataset are generated by the algorithm proposed in the paper [1]; the label distributions of Yeast and Human-Gene datasets are obtained from the biological laboratory equipments; s-JAFFE and SBU-3DFE datasets have label distributions with all description degree values greater than zero, which will produce rankings over the whole label set (i.e., label ranking) instead of multi-label rankings. Therefore, we only chosen the Movie dataset from Geng's 15 datasets.
>
> [1] Geng, X. and L. Luo. Multilabel Ranking with Inconsistent Rankers. CVPR (2014): 3742-3747.
>
> **Q4: Authors points out two limitations in Section 6. For the 2nd one, i.e., "If several labels actually describe the instance to the same degree, then requiring experts to give the strict order of these labels may lead to errors and invalidate Theorem 1", authors should extend the proposed framework to deal with such kind of data sets, not treat it as a future work. After all, it is more realistic that some labels have the same description degree.**
>
> A: We did not consider the situation of "identical label discription degree" in this paper because the analytic form of EAE and its corollaries are too complex to be shown together with the rest of the paper in a limited pages. What is more, this situation may only invalidate Theorem 1 and its corollaries. For our proposed DRAM framework, as stated in the conclusion of our paper, only minor modifications are needed to accommodate this situation. For example, given a label ranking $y_1<y_2=y_3$, where $y_i$ denotes the $i$-th label, when generating $\\boldsymbol z$ (i.e., line 5 of Algorithm 1), we only need to generate a random real-valued vector for $y_1<y_2$, say $[0.3, 0.5]$, and set the corresponding value of $y_3$ to the value of $y_2$, and end up with a vector $[0.3,0.5,0.5] $, and finally normalize this vector to obtain a label distribution.

---

### Official Review · Reviewer_YEUF · 2022-07-05

**Rating:** 5
**Confidence:** 4
**Soundness:** 2 fair
**Presentation:** 2 fair
**Contribution:** 3 good

**Summary:**

This paper studies how to exploit multi-label ranking, a compromise w.r.t. annotation cost, to learn a predictive model on label distribution. The authors theoretically investigate the relation between multi-label ranking and label distribution and demonstrate its superior over the logical labels. Then a general framework involving custom knowledge is proposed to recover and learn label distribution end-to-end. The extensive experiments show the effectiveness of the proposal.

**Questions:**

1. What is the time cost of the EM algorithm used to train model? The authors should add more analysis on time complexity.
2. Note that the datasets used in this paper are relatively small, so what is the capability of the proposed framework on large datasets?
3. Are there any selection criteria for scoring function? Note that only DRAM+LN is tested on all datasets and how do different scoring functions affect the results?
4. Multi-label ranking may still require a high cost. The authors can add some discussion on cost of various methods (the proposal, LE and LDL methods), for example, the cost comparison when achieving the same performance.


**Limitations:**

Yes

**Strengths And Weaknesses:**

This is the first paper that proposes to predict label distribution from the view of multi-label ranking and shows the good guarantees for performance. The originality is good and the main idea of this paper is clearly presented. They also show better results than other compared methods. Despite that, I still have some concerns:
1. What is the time cost of the EM algorithm used to train model? The authors should add more analysis on time complexity.
2. Note that the datasets used in this paper are relatively small, so what is the capability of the proposed framework on large datasets? In fact, label ranking is often associated with high time complexity.
3. Are there any selection criteria for scoring function? Note that only DRAM+LN is tested on all datasets and how do different scoring functions affect the results?
4. Multi-label ranking may still require a high cost. The authors can add some discussion on cost of various methods (the proposal, LE and LDL methods), for example, the cost comparison when achieving the same performance.
5. Some spelling mistakes need to be corrected, e.g., “internel” in the third sentence in section 2.1 should be “internal”.

---

> ### Author Response · Authors · 2022-07-29
> **Response to Reviewer YEUF**
>
> Many thanks for your comments, we have provided point-by-point responses to your questions below.
>
> **Q1: What is the time cost of the EM algorithm used to train model?**
>
> A: The complexity of the training procedure for DRAM is dominated by the E-step and M-step, and depends on the basic learner used. Each EM iteration in DRAM takes $O((LM+TP)KN)$, where $K$, is the number of Dirichlet components in the mixture, $N$ is the number of observations, $P$ is the number of learnable parameters of the used basic learner (for example, for DRAM+LN, $P$ is equal to the number of feature variables multiplied by the number of labels), $M$ is the number of labels, $L$ is the number of Monte Carlo samples, $T$ is the number of iterations required for the M-step to reach convergence.
>
> **Q2: what is the capability of the proposed framework on large datasets?**
>
> A: Most label ranking methods have high complexity due to the pairwise or listwise operations involved. In our paper, we avoid these operations (and thus avoiding the high time complexity) by converting label ranking into a random vector, and thus our framework also works on large-scale datasets.
>
> **Q3: Are there any selection criteria for scoring function? Note that only DRAM+LN is tested on all datasets and how do different scoring functions affect the results?**
>
> A: Since our paper is not concerned with designing a good scoring function, we follow the principle of "the simpler the better" and choose the non-informative scoring function. Of course, most existing LE assumptions can be used as a scoring function with simple modifications. For example, a common assumption of LE is that semantically similar labels are closer w.r.t. label description degree (label correlation assumption). We can define $\\phi(d)={(\\sum_{i,j}|(d_i-d_j)^2-s_{ij}|)^{-1}}$  as the scoring function to encode this assumption, where $s_{ij}$ is the normalized distance between the i-th and j-th column of permutation matrix. The experimental results of using this scoring function are as follows (all hyperparameters keep unchanged). Each entry in the table represents "DRAM+LC/DRAM+LN", where DRAM+LC is DRAM with the new scoring function.
>
> | Dataset| Cheb|Canber|Cosine|Rho|
> |:--|-|-|-|-|
> |Movie|0.123/0.124|1.055/1.058|0.934/0.932|0.722/0.720|
> |Emotion6|0.281/0.282|3.956/3.953|0.783/0.785 | 0.592/0.588 |
> |Twitter-LDL|0.355/0.355|6.525/6.526|0.828/0.828 | 0.601/0.604 |
> |Flickr-LDL|0.324/0.324|6.015/6.013|0.814/0.815 | 0.625/0.627 |
> |NSRD-e1|0.517/0.509|7.655/7.649|0.591/0.599 | 0.453/0.459 |
> |NSRD-e2|0.517/0.509|7.655/7.649|0.592/0.599 | 0.453/0.459 |
> |NSRD-e3|0.559/0.554|7.699/7.699|0.574/0.577 | 0.451/0.455 |
>
> It can be seen that the new scoring functions have a small effect on results. For time reasons, the new scoring functions are only the simplest implementation of label correlation assumption; we believe that a well-designed scoring function can effectively improve the performance.
>
> **Q4: Multi-label ranking may still require a high cost. The authors can add some discussion on cost of various methods (the proposal, LE and LDL methods), for example, the cost comparison when achieving the same performance.**
>
> A: (1) From the empirical perspective, logical label annotating requires giving whether each label is relevant to the instance; multi-label ranking requires additionally giving the ranking of relevant labels; label distribution requires further answering the question of how preferable is one label to another label. Obviously, w.r.t. annotating cost, logical label is the smallest, multi-label ranking is moderate, and label distribution is the largest. As for the quantitative difference in these costs, we cannot give precise results because we did not record the exact time of annotating. Nevertheless, we can give a rough conclusion according to our experience in annotating the NSRD dataset. During creating the NSRD, the ratio of time we spent on annotating with logical label, label ranking, and label distribution was about 1:2:4.
>
> (2) From the theoretical perspective, label distribution annotating produces an EAE (expected approximation error) of zero. According to the EAE formulas for multi-label ranking and logical label (as shown in Eq. (2) and Eq. (5) in our paper), multi-label ranking and logical label can theoretically never achieve the same performance as label distribution annotation. Nevertheless, by observing the EAE at different values of m, we can get an idea of how many labels need to be annotated to reach a certain error value:
>
> |m|1|2|3|4|5| 6|7|8|9|10|
> |-|-|-|-|-|-|-|-|-|-|-|
> |$E_\\sigma$ |0.08| 0.07|0.067|0.065|0.063|0.062|0.061|0.06|0.06|0.06|
> |$E_l$| 0.097|0.215|0.366|0.524|0.686|0.85|1.014|1.179|1.344|1.51|
>
> where $E_{\\sigma}$ is the expected EAE defined in Corollary 3 of our paper, and ${E}_{ l}$ is the expected value of EAE for logical label when $\\delta\\sim\\text{Uni}(0,m^{-1})$ and $\\hat\\delta\\sim \\text{Uni}(0, m^{-1})$.

---

### Official Review · Reviewer_KFoB · 2022-07-07

**Rating:** 6
**Confidence:** 4
**Soundness:** 3 good
**Presentation:** 3 good
**Contribution:** 2 fair

**Summary:**

This article propose a generic framework named DRAM(label Distribution predicting from
multi-label RAnking via conditional Dirichlet Mixtures). It allows to flexibly encode the prior knowledge about the tasks by a scoring function, and it integrates the processes of recovering and learning label distributions end-to-end. Besides, the author theoretically investigate the relation between multi-label ranking and label distribution and define the notion of EAE to quantify the quality of an annotation, and give the bounds of EAE for multi-label ranking.


**Questions:**

2. It can be seen from Corollary 1 that the optimal solution of  ${\hat \delta ^*} = ((2m + 1)\delta  + m){(m + 1)^{ - 2}}$ does not approach  ${\delta }$, whether this shows the capability boundary of the method? In addition, the process of theoretically guiding the algorithm is mainly reflected in the update of ${{\hat \delta } _n} = \left| {{\sigma _n}} \right|{(\left| {{\sigma _n}} \right| + 1)^{ - 2}}$. From the Corollary 3, ${\hat \delta }  \in [m{(m + 1)^{ - 2}},{m^{ - 1}}]$, why does  ${\delta }$ take the lower bound of the interval instead of a certain value in the middle ?


**Limitations:**

3. When analyzing how the number of mixture components K affects the performance of the method, the authors point out that “t can be seen that the performance gets better first and then worse as K increases.”, it seems that such a conclusion cannot be simply drawn from the figure.

**Strengths And Weaknesses:**

Generally speaking, the article has a certain degree of innovation and the theoretical results are plentiful. The structure of the article is complete and the thinking is clear. Therefore, we hold the opinion of weak reception.
1. In Definition 1, the author states that $z$ and $\hat z$ are independent, but from Corollary 1 it can be seen that ${\delta }$ and ${\hat \delta }$ are closely related, so the independence of  $z$ and $\hat z$ needs further explanation.

---

> ### Author Response · Authors · 2022-07-29
> **Response to Reviewer KFoB**
>
> Many thanks for your precious comments and corrections. we have provided point-by-point responses to your questions below.
>
> **Q1: In Definition 1, the author states that $z$ and $\\hat z$ are independent, but from Corollary 1 it can be seen that $\\delta$ and $\\hat \\delta$ are closely related, so the independence of $z$ and $\\hat z$ needs further explanation.**
>
> A: If we know the value of $\\delta$, then Corollary 1 can tell us what $p(\\hat z)$ minimizes the EAE, which means that $\\delta$ can affect $p(\\hat z)$. But, the independence of $z$ and $\\hat z$ is determined by $p(z,\\hat z)=p(z)p(\\hat z)$, and $\\delta$ clearly does not affect this equality. What is more, the value of $\\delta$ is actually unknown. As shown in line 133 of our paper, the specific value of $\\hat \\delta$ is taken from $[m(m+1)^{-2},m^{-1}]$, which is not directly related to $\\delta$.
>
> **Q2: It can be seen from Corollary 1 that the optimal solution of $\\hat\\delta^\\star=((2m+1)\\delta+m)(m+1)^{-2}$ does not approach $\\delta$, whether this shows the capability boundary of the method?**
>
> A: Yes, the bound of EAE is shown in Corollary 2 of our paper.
>
> **Q3: In addition, the process of theoretically guiding the algorithm is mainly reflected in the update of $\\hat \\delta_n=|\\sigma_n|(|\\sigma_n|+1)^{-2}$. From the Corollary 3, $\\hat\\delta\\in[m(m+1)^{-2}, m^{-1}]$, why does $\\delta$ take the lower bound of the interval instead of a certain value in the middle?**
>
> A: The parameter $\\hat\\delta$ can determine the range of label distribution (i.e., $\\mathcal S_{\\boldsymbol\\sigma}^{\\hat\\delta}$), and the volume of $\\mathcal S_{\\boldsymbol\\sigma}^{\\hat\\delta}$ equals ${(1-m\\hat\\delta)^m}{(m!)^{-1}}$ as shown in the line 14 of the Appendix. This means that the volume of the label distribution's space is smaller when $\\hat \\delta$ is larger. If the range of label distribution is too small, the label distribution will tend to be a regular step shape (e.g., $[0, 0.1,0.2,0.3,0.4]$), which will result in the scoring function being useless, and thus unable to portray the relative importance information among the labels. Therefore, we take the lower bound of $\\hat\\delta$.
>
> **Q4: When analyzing how the number of mixture components K affects the performance of the method, the authors point out that “it can be seen that the performance gets better first and then worse as K increases.”, it seems that such a conclusion cannot be simply drawn from the figure.**
>
> A: Many thanks for your correction. This conclusion is indeed somewhat arbitrary, and we consider revising it to "It can be seen that appropriately increasing the Dirichlet components in the mixture can improve the model capacity and thus improve the predictive performance, but too many Dirichlet components may lead to overfitting and thus degrade the predictive performance.".

---

### Official Review · Reviewer_rKfn · 2022-07-11

**Rating:** 5
**Confidence:** 4
**Soundness:** 2 fair
**Presentation:** 3 good
**Contribution:** 3 good

**Summary:**

This paper studies Label Enhancement (LE). Instead of enhancing label distribution from logical labels, it proposes to recovery label distribution from multi-label ranking annotations. To achieve that, the authors establish several theories and put forward a new method. Experimental results validate the advantages of recovering label distribution from ranking over logical labels.

**Questions:**

 I expect to see the results in terms of KL divergence in the rebuttal.

**Limitations:**

I believe the authors have clarified the limitations of their paper.

**Strengths And Weaknesses:**

On one hand, the strengths include:
+ LE with ranking annotation is original, which hasn't been noticed in the field of LE. Besides, the authors theoretically prove that LE from ranking is better than LE from logical labels, as shown in corollary 4.
+ The method DRAM is novel. DRAM uses the Dirichlet mixtures and EM algorithm to recover label distribution from rankings.
+ Most importantly, the experiments show that DRAM has remarkable improvements over existing LE (from logical labels). Moreover, the DRAM+LN combination even outperforms SA and DM with the ground-truth label distributions, which is impressive.

On the other hand, for me, the main weakness mainly lies in that the paper is not easy to follow. I understand that the authors have put necessary proofs and details to the supplementary material, but still the Sections 2 and 3 are not easily understandable. Besides, I expect to see the results in terms of KL divergence because it is such an important measure metric for LDL.

---

> ### Author Response · Authors · 2022-07-29
> **Response to Reviewer rKfn**
>
> Thank you for your valuable comments, we have provided point-by-point responses to your questions below.
>
> **Q1: On the other hand, for me, the main weakness mainly lies in that the paper is not easy to follow. I understand that the authors have put necessary proofs and details to the supplementary material, but still the Sections 2 and 3 are not easily understandable.**
>
> A: We will make some necessary changes to make Sections 2 and 3 easier to understand.
>
> **Q2: Besides, I expect to see the results in terms of KL divergence because it is such an important measure metric for LDL.**
>
> A: The performance of the comparison algorithms on KL divergence is shown in the following table, where the notation meaning is consistent with Table 1 in our paper.
>
> | Method   | Movie                         | Emotion6                      | Twitter-LDL                   | Flickr-LDL                    |
> | :------- | :---------------------------- | :---------------------------- | :---------------------------- | :---------------------------- |
> | DRAM+LN  | $(2)\\ 0.104\\pm0.002$          | $(1)\\ 0.488\\pm0.014$          | $(1)\\ 0.696\\pm0.025$          | $(1)\\ 0.623\\pm0.009$          |
> | DT+VI+SA | $(6)\\ 0.163\\pm0.002\\ \\bullet$ | $(3)\\ 0.586\\pm0.029\\ \\bullet$ | $(6)\\ 1.162\\pm0.009\\ \\bullet$ | $(6)\\ 0.986\\pm0.013\\ \\bullet$ |
> | DT+VI+DM | $(7)\\ 0.163\\pm0.002\\ \\bullet$ | $(2)\\ 0.571\\pm0.021\\ \\bullet$ | $(5)\\ 1.156\\pm0.014\\ \\bullet$ | $(5)\\ 0.982\\pm0.014\\ \\bullet$ |
> | DT+GL+SA | $(4)\\ 0.133\\pm0.001\\ \\bullet$ | $(5)\\ 0.696\\pm0.027\\ \\bullet$ | $(3)\\ 1.059\\pm0.004\\ \\bullet$ | $(3)\\ 0.933\\pm0.009\\ \\bullet$ |
> | DT+GL+DM | $(3)\\ 0.130\\pm0.001\\ \\bullet$ | $(4)\\ 0.653\\pm0.019\\ \\bullet$ | $(4)\\ 1.105\\pm0.005\\ \\bullet$ | $(4)\\ 0.960\\pm0.009\\ \\bullet$ |
> | GT+SA    | $(5)\\ 0.161\\pm0.007\\ \\bullet$ | $(7)\\ 3.229\\pm0.218\\ \\bullet$ | $(7)\\ 1.983\\pm0.221\\ \\bullet$ | $(7)\\ 1.665\\pm0.100\\ \\bullet$ |
> | GT+DM    | $(1)\\ 0.098\\pm0.002\\ \\circ$   | $(6)\\ 1.034\\pm0.049\\ \\bullet$ | $(2)\\ 0.912\\pm0.024\\ \\bullet$ | $(2)\\ 0.925\\pm0.014\\ \\bullet$ |
>
> | Method   | NSRD-e1                       | NSRD-e2                       | NSRD-e3                       |
> | :------- | :---------------------------- | :---------------------------- | :---------------------------- |
> | DRAM+LN  | $(3)\\ 1.222\\pm0.018$          | $(3)\\ 1.219\\pm0.021$          | $(3)\\ 1.279\\pm0.035$          |
> | DT+VI+SA | $(7)\\ 1.582\\pm0.042\\ \\bullet$ | $(7)\\ 1.557\\pm0.051\\ \\bullet$ | $(7)\\ 1.599\\pm0.058\\ \\bullet$ |
> | DT+VI+DM | $(6)\\ 1.510\\pm0.037\\ \\bullet$ | $(6)\\ 1.488\\pm0.031\\ \\bullet$ | $(6)\\ 1.562\\pm0.038\\ \\bullet$ |
> | DT+GL+SA | $(5)\\ 1.436\\pm0.014\\ \\bullet$ | $(5)\\ 1.425\\pm0.012\\ \\bullet$ | $(5)\\ 1.486\\pm0.012\\ \\bullet$ |
> | DT+GL+DM | $(4)\\ 1.427\\pm0.016\\ \\bullet$ | $(4)\\ 1.419\\pm0.012\\ \\bullet$ | $(4)\\ 1.480\\pm0.014\\ \\bullet$ |
> | GT+SA    | $(1)\\ 1.179\\pm0.037\\ \\circ$   | $(1)\\ 1.149\\pm0.044\\ \\circ$   | $(2)\\ 1.200\\pm0.053\\ \\circ$   |
> | GT+DM    | $(2)\\ 1.190\\pm0.044\\ \\circ$   | $(2)\\ 1.187\\pm0.039\\ \\circ$   | $(1)\\ 1.187\\pm0.047\\ \\circ$   |
>
> By observing this table above and Table 1 in our paper, it can be found that the experimental results on KL divergence are very similar to those on the Cheb metric. Therefore, in consideration of page limitation, we do not show the KL divergence.

---

### Meta-Review · Area_Chair_TK1W · 2022-08-28

**Recommendation:** Accept
**Confidence:** Less certain

**Metareview:**

This paper studies the problem of predicting label distribution from multi-label ranking. First, the authors give a theoretical analysis to prove the superiority of the multi-label ranking over the logical labels. Then an end-to-end framework called DRAM is proposed for recovering and learning label distributions. The corresponding experiments validate the effectiveness of the proposed algorithms. Overall, this work is technically solid. The concerns raised by reviewers are not so serious and have been answered. I recommend accepting this paper.

**Award:**

No

---

### Decision · Program_Chairs · 2022-09-14

Accept